# TOWARDS UNDERSTANDING MEMORY BUFFER BASED CONTINUAL LEARNING

## ABSTRACT

Continual learning (CL) is a paradigm that adapts to and retains knowledge from a stream of tasks. Despite the growing number of experimental methods in CL, there is a lack of rigorous theoretical analysis, particularly in memory-based continual learning (MCL), which remains an open research area. In this paper, we theoretically analyze the impact of memory in CL and derive explicit expressions for expected forgetting and generalization errors under overparameterized linear models. We propose a detailed matrix decomposition of the data to distinguish between current and previous datasets, effectively decoupling the coupled data for different tasks. Additionally, we conduct a comprehensive mathematical analysis for scenarios with a small number of tasks and employ numerical analysis for larger task scenarios to evaluate the overall properties of expected forgetting and generalization errors. Compared with CL, our theoretical analysis suggests that (1) a larger memory buffer must be paired with a larger model to effectively reduce forgetting; (2) training with a larger memory buffer generalizes better when tasks are similar but may perform worse when tasks are dissimilar, while training with a large model can help mitigate this negative effect. Ultimately, our findings here sheds new light on how memory can assist CL in mitigating catastrophic forgetting and improving generalization.

## 1 INTRODUCTION

Continual learning (CL) aims to dynamically learn from a sequence of tasks without forgetting previously acquired knowledge. Traditional neural networks are typically trained on fixed datasets and evaluated in static environments. However, in real-world scenarios, data is often presented in a non-stationary manner, requiring models to update their knowledge continuously. This dynamic learning faces a major challenge known as catastrophic forgetting (McCloskey & Cohen, 1989), where acquiring new information results in the forgetting of previously learned knowledge.

Even though various empirical continual learning (CL) methods (Riemer et al., 2018; Buzzega et al., 2020; Bohao et al., 2024) have been proposed recently, rigorous theoretical analysis remains limited, particularly for memory buffer-based continual learning (MCL). As MCL is one of the most straightforward and widely used approaches to mitigate catastrophic forgetting, a thorough theoretical investigation into its mechanisms and limitations is essential. Recent works have attempted to develop a theoretical understanding of memory in CL (Knoblauch et al., 2020; Chen et al., 2022; Han et al., 2023). However, none of these studies provide explicit forms to characterize forgetting and generalization for MCL. In this paper, we consider both limited and unlimited-size memory buffers to provide a more comprehensive analysis of memory buffers. We derive explicit expressions under overparameterized linear models to offer a clearer understanding of how memory can assist CL in mitigating catastrophic forgetting and improving generalization.

To conduct these theoretical studies, there are two main challenges: (i) The training dataset contains both the data from previous and current tasks, resulting in the coupling of the data matrix and label vector. This coupling means that it is not possible to represent the labels for each task using a single ground truth matrix. (ii) In the expressions for forgetting and generalization error, the increasing and decreasing terms are coupled, making it difficult to analyze their monotonicity directly. To address these challenges, we first perform a detailed matrix decomposition of the data matrix and the label vector, using the ground truth matrix from each previous task in combination to represent

**Table 1:** Comparison of reservoir sampling-based memory buffer (RMB) and full rehearsal memory buffer (FMB) with no memory buffer (NMB). Here, $T$ represents the number of training tasks, $s$ denotes the number of samples per task, $d$ represents the number of training model parameters, and $M_{max}$ is the maximum size of the memory buffer. To simplify notation, we define $r := 1 - \frac{s}{d}$, $c_{i,j} := r^{T-i} + r^{T-j} - r^{j-i}$, $u := 1 - \frac{M_{max}+s}{d}$, $u_{i,j} := u^{T-i} + u^{T-j} - u^{j-i}$, and $f(l) := 1 - \frac{ls}{d}$.

| TYPE OF MEMORY BUFFER | FORGETTING & GENERALIZATION ERROR | MEMORY |
|---|---|---|
| NMB (LIN ET AL., 2023) | $\mathbb{E}[F_T] = \frac{1}{T-1}\sum_{i=1}^{T-1}\left[(r^T - r^i)\|\boldsymbol{w}_i^*\|^2 + \frac{s}{d}\sum_{j>i}^{T}c_{i,j}\|\boldsymbol{w}_j^* - \boldsymbol{w}_i^*\|^2\right]$ | ✗ |
| | $\mathbb{E}[G_t] = \frac{1}{T}\sum_{i=1}^{T}r^T\|\boldsymbol{w}_i^*\|^2 + \frac{s}{Td}\sum_{i=1}^{T}\sum_{k=1}^{T}r^{T-i}\|\boldsymbol{w}_k^* - \boldsymbol{w}_i^*\|^2$ | ✗ |
| RMB (OURS) | $\mathbb{E}[F_T] = \frac{1}{T-1}\sum_{i=1}^{T-1}\left\{\left[(u^{T-1}-u^{i-1})r\|\boldsymbol{w}_i^*\|^2 + \frac{s}{d}\|\boldsymbol{w}_T^* - \boldsymbol{w}_i^*\|^2\right]\right.$ $\left.+\sum_{j>i}^{T-1}\left[\left(\sum_{k=j+1}^{T}+\sum_{k=i+1}^{T}\right)u^{T-k}\frac{M_{max}}{(k-1)d} - \sum_{k=i+1}^{j}u^{j-k}\frac{M_{max}}{(k-1)d} + \frac{u_{i,j}s}{d}\|\boldsymbol{w}_j^* - \boldsymbol{w}_i^*\|^2\right]\right\}$ | ✓ |
| | $\mathbb{E}[G_T] = \frac{1}{T}\sum_{i=1}^{T}\left[u^{T-1}r\|\boldsymbol{w}_T^*\|^2 + \frac{s}{d}\|\boldsymbol{w}_T^* - \boldsymbol{w}_i^*\|\right]$ $+\frac{1}{T}\sum_{i=1}^{T}\sum_{j=1}^{T-1}\left[\sum_{k=j+1}^{T}\frac{u^{T-k}M_{max}}{(k-1)d} + \frac{u^{T-j}s}{d}\right]\|\boldsymbol{w}_j^* - \boldsymbol{w}_i^*\|^2$ | ✓ |
| FMB (OURS) | $\mathbb{E}[F_T] = \frac{1}{T-1}\left\{\sum_{i=1}^{T-1}\left[\prod_{l=1}^{T}f(l) - \prod_{l=1}^{i}f(l)\right]\|\boldsymbol{w}_i^*\|^2\right.$ $\left.+\frac{s}{d}\sum_{j>i}^{T}\left[\left(\sum_{k=j}^{T}+\sum_{k=i}^{T}\right)\prod_{l=k+1}^{T}f(l) - \sum_{k=i}^{j}\prod_{l=k+1}^{j}f(l)\right]\|\boldsymbol{w}_j^* - \boldsymbol{w}_i^*\|^2\right\}$ | ✓ |
| | $\mathbb{E}[G_t] = \frac{1}{T}\sum_{i=1}^{T}\prod_{l=1}^{T}f(l)\|\boldsymbol{w}_i^*\|^2 + \frac{s}{Td}\sum_{i=1}^{T}\sum_{j=1}^{T}\sum_{k=j}^{T}\prod_{l=k+1}^{T}f(l)\|\boldsymbol{w}_j^* - \boldsymbol{w}_i^*\|^2$ | ✓ |

the corresponding label vector. Then, we conduct a comprehensive mathematical analysis for the expression of forgetting and generalization error with a small number of tasks and employ numerical analysis for larger task scenarios.

Here are our insights based on the theoretical results for both reservoir sampling-based memory buffer and full rehearsal memory buffer. For the reservoir sampling-based memory buffer 1) When $T = 2$, increasing the memory buffer size consistently reduces forgetting. However, for $T > 2$, forgetting may worsen unless the condition $\frac{M_{max}+s}{d} < \frac{1}{T-1}$ is satisfied, which ensures the effectiveness of a larger memory buffer; 2) When $T = 2$, a larger memory buffer improves generalization when tasks are highly similar, but it degrades generalization when tasks are highly dissimilar. For $T > 2$, even with highly dissimilar tasks, increasing the memory buffer can still reduce forgetting in certain specific cases. For the full rehearsal memory buffer: 1) when $T = 2$, incorporating a memory buffer effectively mitigates forgetting compared to methods without memory. When $T > 2$, the condition $\frac{s}{d} < \frac{1}{T^2}$ is necessary to ensure good performance. If new tasks are similar to all previous ones, storing more samples leads to less forgetting. 2) In terms of generalization, if all tasks are similar to each other, full rehearsal results in better generalization than training without memory. Conversely, if tasks are highly dissimilar, full rehearsal leads to worse generalization compared to methods without memory.

Our main contributions are summarized as follows.

- We consider the reservoir sampling-based memory buffer as our limited memory buffer and the full rehearsal memory buffer as the unlimited memory buffer. For both buffer types, we derive explicit expressions for the expected forgetting and generalization errors under overparameterized linear models.

- We investigate the impact of memory buffer size and the number of parameters on both forgetting and generalization when training with a reservoir sampling-based memory buffer, revealing the following important insights: (1) A larger memory buffer must be paired with a larger model to reduce forgetting effectively. (2) A sufficiently large model can result in zero forgetting. (3) A larger memory buffer may improve generalization when tasks are highly similar but can degrade generalization when tasks are highly dissimilar.

- We analyze the effects of memory buffer size and model parameters on both forgetting and generalization when training with a full rehearsal memory buffer, revealing: (1) More parameters help mitigate forgetting when tasks are highly similar. (2) A sufficiently large model can lead to zero forgetting. (3) Storing more samples improves generalization when tasks are highly similar but may worsens it when tasks are highly dissimilar.

## 2 RELATED WORK

**Memory buffer-based CL.** Memory buffer-based methods are mainly divided into experience replay and generative replay. Experience replay stores a subset of data from previous tasks in a replay buffer, following specific sample selection principles, and uses it for training in the current task. One common buffer strategy is reservoir sampling (Riemer et al., 2018; Buzzega et al., 2020), where every previously seen example has an equal probability of being stored in the buffer at each training round. More advanced strategies include gradient-based (Saha et al., 2021; Aljundi et al., 2019; Borsos et al., 2020) and optimization-constrained approaches (Chaudhry et al., 2019; Yin et al., 2020; Yoon et al., 2021). Generative replay typically requires an additional generative model to produce synthetic samples for previous tasks. Earlier methods generally use GAN or VAE as the generator (Shin et al., 2017; Chenshen et al., 2018; Zhai et al., 2019). Recently, leveraging the powerful generative capabilities of diffusion models, some methods have begun to explore using diffusion models as the generator (Gao & Liu, 2023; Kim et al., 2024). In addition, there are some methods combine memory with other techniques like knowledge distillation. For example, iCaRL (Rebuffi et al., 2017) uses a buffer to train a nearest-mean-of-exemplars classifier while preventing representation degradation in later tasks through a self-distillation loss.

**Theoretical analysis of CL.** There have been only a few attempts to develop a theoretical understanding of memory in CL. For instance, (Knoblauch et al., 2020) uses set theory to show that achieving perfect memory for optimal CL is NP-hard. (Chen et al., 2022) applies the PAC framework to analyze the lower bound of memory requirements in CL. (Han et al., 2023) provides a convergence analysis of memory-based CL with stochastic gradient descent, framing it as a smooth nonconvex finite-sum optimization problem. Several theoretical studies focus on linear regression tasks. (Evron et al., 2022) proves an upper bound for catastrophic forgetting under specific task orderings. (Lin et al., 2023) presents explicit expressions for forgetting and generalization error in overparameterized linear models. (Goldfarb & Hand, 2023) analyze the impact of overparameterization for linear models in a two-task scenario. (Ding et al., 2024) offers a general theoretical analysis of forgetting in the linear regression model using stochastic gradient descent. (Zhao et al., 2024) provides a statistical analysis of regularization-based continual learning across a sequence of linear regression tasks. However, none of these works provide explicit forms of forgetting and generalization errors for memory-based continual learning (MCL). In contrast, we offer explicit results for MCL without making assumptions about the ground truth vectors and provide a comprehensive discussion on how memory buffer size and the number of parameters affect forgetting and generalization.

The most relevant study to our work is (Lin et al., 2023), which also investigated CL in overparameterized linear models. However, our work differs significantly from (Lin et al., 2023) in the following aspects: (1) We focus on analyzing forgetting and generalization errors when training with memory buffers, whereas (Lin et al., 2023) considered training without memory buffers; (2) Our analysis explores the interplay between memory buffer size, overparameterization, and the number of tasks, emphasizing the conditions under which these factors can mitigate forgetting and enhance generalization. In contrast, (Lin et al., 2023) did not address the impact of the number of tasks; (3) Our findings provide new insights into how memory can assist CL in alleviating catastrophic forgetting and improving generalization.

## 3 PRELIMINARY

### 3.1 PROBLEM SETUP

We consider the standard CL setup with $T$ training tasks. We define $\mathcal{D}_t = \{\boldsymbol{X}_t, \boldsymbol{y}_t\}$ as the training dataset for task $t$, where $\boldsymbol{X}_t \in \mathbb{R}^{d \times s_t}$ is the feature matrix containing $s_t$ samples of $d$-dimensional feature vectors, and $\boldsymbol{y}_t \in \mathbb{R}^{s_t}$ is the corresponding output vector. Additionally, we assume that each element of the feature matrix $\boldsymbol{X}_t$ follows standard Gaussian distribution $\mathcal{N}(0, 1)$ and is independent of each other. For any task $t$, we define a linear ground truth vector $\boldsymbol{w}_t^*$ (Belkin et al., 2018; Evron et al., 2022; Lin et al., 2023), which satisfies $\boldsymbol{y}_t = \boldsymbol{X}_t^T \boldsymbol{w}_t^*$. The set of ground truth vectors for all $T$ tasks is denoted as $\mathcal{W}_T = \{\boldsymbol{w}_1^*, \ldots, \boldsymbol{w}_T^*\}$. This paper focuses on two memory buffer strategies: the reservoir sampling-based memory buffer and the full rehearsal memory buffer. We assume the current size of memory buffer $\mathcal{M}_t$ for the $t$-th task is $M_t$, with a maximum size of $M_{max}$. We define the feature matrix stored in the memory buffer as $\boldsymbol{X}_{\mathcal{M}_t} \in \mathbb{R}^{d \times M_t}$, and the corresponding label vector

as $\boldsymbol{y}_{\mathcal{M}_t} \in \mathbb{R}^{M_t}$. We use $\boldsymbol{e}_n^i \in \mathbb{R}^n$ to denote the standard basis vector of length $n$, where the $i$-th element is 1, and all other elements are 0. Similarly, we define $\boldsymbol{E}_n^i = \mathrm{diag}\{\boldsymbol{e}_n^i\} \in \mathbb{R}^{n \times n}$, where diag creates a diagonal matrix with $\boldsymbol{e}_n^i$ as its diagonal. To simplify subsequent discussions, we set $m_t = \lfloor M_{max}/t \rfloor$, where $\lfloor \cdot \rfloor$ denotes the floor function, and $\bar{m}_t = M_{max} - (t-1)m_t$.

For ease of the statements, we assume the number of samples for each task is equal, as shown in Assumption 1, and without this assumption, our conclusions will not be fundamentally affected.

**Assumption 1.** $s_t = s$ for all $t \in [T]$.

### 3.2 MEMORY BUFFER

#### 3.2.1 RESERVOIR SAMPLING-BASED MEMORY BUFFER

This memory buffer updates using reservoir sampling, ensuring that for task $t$, the probability of any example from the previous $t-1$ tasks being in the buffer is equal. Therefore, **we assume the number of samples stored in the memory buffer for each previous task is nearly equal.** This approach ensures diversity and balance among samples from previous tasks while adhering to the constraints of a limited-size buffer. We limit the buffer capacity to not exceed $s$ and define it as follows.

When $t \geq 2$, the memory buffer stores $m_{t-1}$ samples for each of the previous $t-1$ tasks and 1 sample for each of the $\bar{m}_{t-1}$ distinct tasks from the previous $t-1$ tasks. This ensures that the difference in the number of samples stored in the buffer for each task does not exceed 1, satisfying the above assumption. We assume that each element of $\boldsymbol{X}_{\mathcal{M}_t}$ follows standard Gaussian distribution $\mathcal{N}(0,1)$ and is independent of each other. We generate the corresponding label $\boldsymbol{y}_{\mathcal{M}_t} \in \mathbb{R}^{M_{max}}$ as follows:

$$\boldsymbol{y}_{\mathcal{M}_t} = \sum_{k=1}^{t-1} \sum_{l=(k-1)m_{t-1}+1}^{km_{t-1}} \boldsymbol{E}_{M_{max}}^l \boldsymbol{X}_{\mathcal{M}_t}^\top \boldsymbol{w}_k^* + \sum_{j=1}^{\bar{m}_{t-1}} \boldsymbol{E}_{M_{max}}^{(t-1)m_{t-1}+j} \boldsymbol{X}_{\mathcal{M}_t}^\top \boldsymbol{w}_{t_j}^*, \tag{1}$$

where $\{\boldsymbol{w}_{t_1}^*, \ldots, \boldsymbol{w}_{t_{\bar{m}_{t-1}}}^*\} \subseteq \mathcal{W}_t$, with each element uniformly drawn from $\mathcal{W}_t$.

#### 3.2.2 FULL REHEARSAL MEMORY BUFFER

A full rehearsal memory buffer stores all samples from previous tasks. Training a model with this buffer means retraining with all samples, including both old and new tasks, making it the most straightforward memory-based approach. However, it requires significant computational resources and memory storage. We define the full rehearsal memory buffer as follows.

When $t \geq 2$, the memory buffer stores all samples from the previous $t-1$ tasks. We assume that each element of $\boldsymbol{X}_{\mathcal{M}_t}$ follows standard Gaussian distribution $\mathcal{N}(0,1)$ and is independent of each other. Therefore, $M_t = (t-1)s$ and we generate the corresponding label $\boldsymbol{y}_{\mathcal{M}_t} \in \mathbb{R}^{M_t}$ as follows:

$$\boldsymbol{y}_{\mathcal{M}_t} = \sum_{k=1}^{t-1} \sum_{l=(k-1)s+1}^{ks} \boldsymbol{E}_{M_t}^l \boldsymbol{X}_{\mathcal{M}_t}^\top \boldsymbol{w}_k^*. \tag{2}$$

### 3.3 TRAINING SETTING

**Training dataset.** For any task $t \in \{2, \ldots, T\}$, we consider training with the original dataset $\boldsymbol{X}_t$ alongside the memory buffer $\mathcal{M}_t$. Thus, we define the expanded training dataset $\hat{\mathcal{D}}_t := \{\hat{\boldsymbol{X}}_t, \hat{\boldsymbol{y}}_t\}$ for task $t$ by concatenating $\mathcal{M}_t$ and $\mathcal{D}_t$, where $\hat{\boldsymbol{X}}_t \in \mathcal{R}^{d \times (s+M_t)}$ is defined as $[\boldsymbol{X}_{\mathcal{M}_t}, \boldsymbol{X}_t]$ and $\hat{\boldsymbol{y}}_t \in \mathcal{R}^{s+M_t}$ is defined as $[\boldsymbol{y}_{\mathcal{M}_t}^\top, \boldsymbol{y}_t^\top]^\top$. To simplify our analysis, we consider the expanded training dataset with $i.i.d.$ Gaussian features, which is stated in the following Assumption 2.

**Assumption 2.** *For any task* $t \in \{2, \ldots, T\}$, *each element of* $\hat{\boldsymbol{X}}_t$ *follows standard Gaussian distribution* $\mathcal{N}(0,1)$ *and is independent of each other.*

**Training procedure.** We train our model parameters sequentially on a series of tasks. To simplify our analysis, we always set the initial training weight for the first task to zero matrix, although setting it to any other fixed matrix does not affect our theoretical results. In each training round, we use

the updated weight from the previous task as the initial weight for the next task. For any task $t$, we consider the mean squared error (MSE) with respect to $\hat{\mathcal{D}}_t$ as our training loss:

$$\mathcal{L}_t^{tr}(\boldsymbol{w}, \hat{\mathcal{D}}_t) = \frac{1}{s}\|\hat{\boldsymbol{X}}_t^\top \boldsymbol{w} - \hat{\boldsymbol{y}}_t\|_2^2. \tag{3}$$

When we focus on the overparameterized regime, there are infinitely many solutions that make Equation (3) equal to zero. We use stochastic gradient descent (SGD) to minimize Equation (3). The convergence point of SGD is the unique solution among these infinitely many solutions, which can be found by solving the following optimization problem:

$$\min_{\boldsymbol{w}_t} \|\boldsymbol{w}_t - \boldsymbol{w}_{t-1}\|_2, \quad s.t. \ \hat{\boldsymbol{X}}_t^\top \boldsymbol{w}_t = \hat{\boldsymbol{y}}_t. \tag{4}$$

We can use the method of Lagrange multipliers to directly obtain the solution to the optimization problem in Equation (4). The resulting solution is given by:

$$\boldsymbol{w}_t = \boldsymbol{w}_{t-1} + \hat{\boldsymbol{X}}_t(\hat{\boldsymbol{X}}_t^\top \hat{\boldsymbol{X}}_t)^{-1}(\hat{\boldsymbol{y}}_t - \hat{\boldsymbol{X}}_t^\top \boldsymbol{w}_{t-1}). \tag{5}$$

## 4 THEORETICAL RESULTS ON FORGETTING AND GENERALIZATION

For the overparameterized model, we define $\mathcal{L}_t(\boldsymbol{w})$ as the model error for task $t \in [T]$:

$$\mathcal{L}_t(\boldsymbol{w}) = \|\boldsymbol{w} - \boldsymbol{w}_t^*\|_2^2. \tag{6}$$

which characterizes the generalization performance of $\boldsymbol{w}$ on task $t$. As shown in (Chaudhry et al., 2019; Lin et al., 2023), we can define forgetting and overall generalization error similarly as follows: (1) *Forgetting*. This metric quantifies how much the model forgets previous tasks after being trained on a sequence of tasks. For any $t \in [2, T]$, we define it as follows:

$$F_t = \frac{1}{t-1}\sum_{i=1}^{t-1}(\mathcal{L}_i(\boldsymbol{w}_t) - \mathcal{L}_i(\boldsymbol{w}_i)). \tag{7}$$

(2) *Overall generalization error*. It assesses the model's generalization performance from the final training round by calculating the average model error across all tasks:

$$G_t = \frac{1}{t}\sum_{i=1}^{t}\mathcal{L}_i(\boldsymbol{w}_T). \tag{8}$$

These are two important metrics for evaluating the performance of continual learning (CL) in the overparameterized regime. To facilitate a comparison with training methods that do not utilize memory, we refer to Lemma 1, which is derived from (Lin et al., 2023) without considering noise. For notational simplicity, we adopt the notation $r := 1 - \frac{s}{d}$ and $c_{i,j} := r^{T-i} + r^{T-j} - r^{j-i}$.

**Lemma 1** (Lin et al. (2023)). *When $d > s + 2$, we have*

$$\mathbb{E}[F_T] = \frac{1}{T-1}\sum_{i=1}^{T-1}\Big[\underbrace{(r^T - r^i)}_{Term\ F_1^1}\|\boldsymbol{w}_i^*\|^2 + \frac{s}{d}\sum_{j>i}^{T}c_{i,j}\|\boldsymbol{w}_j^* - \boldsymbol{w}_i^*\|^2\Big], \tag{9}$$

$$\mathbb{E}[G_t] = \frac{1}{T}\sum_{i=1}^{T}r^T\|\boldsymbol{w}_i^*\|^2 + \frac{s}{Td}\sum_{i=1}^{T}\sum_{k=1}^{T}r^{T-i}\|\boldsymbol{w}_k^* - \boldsymbol{w}_i^*\|^2. \tag{10}$$

Lemma 1 provides the result for the case without a buffer. Next, We consider the impact of memory on CL. We will present explicit forms of forgetting and overall generalization error for memory-based CL using reservoir sampling-based buffers and full rehearsal buffers as follows.

### 4.1 RESERVOIR SAMPLING-BASED MEMORY BUFFER

The reservoir sampling-based memory buffer is a typical limited-sized buffer that ensures every previously seen example has an equal probability of being stored at each training round. To simplify our statements, we define $u := 1 - \frac{M_{max}+s}{d}$, and $u_{i,j} := u^{T-i} + u^{T-j} - u^{j-i}$. Next, we will provide explicit expressions in Theorem 1 for the expected forgetting and generalization errors for CL trained with this buffer under overparameterized linear models.

**Theorem 1.** *Suppose $d > s + M_{max} + 2$. When $T \geq 2$, we have*

$$
\mathbb{E}[F_T] = \frac{1}{T-1} \sum_{i=1}^{T-1} \left\{ \Big[ \underbrace{(u^{T-1} - u^{i-1})r}_{Term\ F_2^1} \|\boldsymbol{w}_i^*\|^2 + \frac{s}{d}\|\boldsymbol{w}_T^* - \boldsymbol{w}_i^*\|^2 \Big] \right.
$$

$$
\left. + \sum_{j>i}^{T-1} \underbrace{\left[ \left( \sum_{k=j+1}^{T} + \sum_{k=i+1}^{T} \right) u^{T-k} \frac{M_{max}}{(k-1)d} - \sum_{k=i+1}^{j} u^{j-k} \frac{M_{max}}{(k-1)d} + \frac{u_{i,j}s}{d} \right]}_{Term\ F_2^2} \|\boldsymbol{w}_j^* - \boldsymbol{w}_i^*\|^2 \right\}
$$

$$\tag{11}$$

$$
\mathbb{E}[G_T] = \frac{1}{T} \sum_{i=1}^{T} \Big[ \underbrace{u^{T-1}r}_{Term\ G_2^1} \|\boldsymbol{w}_i^*\|^2 + \frac{s}{d}\|\boldsymbol{w}_T^* - \boldsymbol{w}_i^*\| \Big]
$$

$$
+ \frac{1}{T} \sum_{i=1}^{T} \sum_{j=1}^{T-1} \underbrace{\left[ \sum_{k=j+1}^{T} \frac{u^{T-k}M_{max}}{(k-1)d} + \frac{u^{T-j}s}{d} \right]}_{Term\ G_2^2} \|\boldsymbol{w}_j^* - \boldsymbol{w}_i^*\|^2
$$

$$\tag{12}$$

*Proof Sketch.* First, we perform a detailed matrix decomposition of the data $\hat{\boldsymbol{X}}_t$:

$$
\sum_{k=1}^{t-1} \sum_{l=(k-1)m_{t-1}+1}^{km_{t-1}} \boldsymbol{E}_{M_{max}+s}^l \hat{\boldsymbol{X}}_t^\top \boldsymbol{w}_k^* + \sum_{j=1}^{\bar{m}_{t-1}} \boldsymbol{E}_{M_{max}+s}^{(t-1)m_{t-1}+j} \hat{\boldsymbol{X}}_t^\top \boldsymbol{w}_{t_j}^* + \sum_{l=M_{max}+1}^{M_{max}+s} \boldsymbol{E}_{M_{max}+s}^l \hat{\boldsymbol{X}}_t^\top \boldsymbol{w}_t^*.
$$

$$\tag{13}$$

Second, we perform a detailed decomposition of the projector onto the row space of $\hat{\boldsymbol{X}}_t$. Then, we provide the corresponding probabilistic properties and algebraic characteristics of the decomposed projector. A comprehensive discussion is provided in Appendix A.1.

Third, we calculate the model error by combining the decomposed data matrix with the decomposed projector matrix obtained in the first two steps:

$$
\mathbb{E}\|\boldsymbol{w}_t - \boldsymbol{w}_i^*\|^2 = u\mathbb{E}\|\boldsymbol{w}_{t-1} - \boldsymbol{w}_i^*\|^2 + \frac{M_{max}}{(t-1)d} \sum_{j=1}^{t-1} \|\boldsymbol{w}_j^* - \boldsymbol{w}_i^*\|^2 + \frac{s}{d}\|\boldsymbol{w}_t^* - \boldsymbol{w}_i^*\|^2. \tag{14}
$$

Finally, we recursively apply Equation (14) and substitute the results into Equation (7) and (8) to obtain explicit expressions for expected forgetting and generalization errors. Then, we finish the proof and leave the details of this proof in Appendix B.1. $\qquad\square$

Before diving into the discussion, we first provide the definition of task similarity in Definition 1.

**Definition 1.** *The task similarity between any tasks $j$ and $i$ is defined by $\|\boldsymbol{w}_j^* - \boldsymbol{w}_i^*\|^2$*

An interesting discovery is that incorporating a memory buffer does not always reduce forgetting or generalization errors. In some cases, it can even exacerbate forgetting, leading to wasted memory and unnecessary computation. Therefore, it is crucial to analyze $\mathbb{E}[F_T]$ and $\mathbb{E}[G_T]$. Specifically, we focus on the coefficients in front of each norm to avoid imposing additional constraints on the norm of the ground truth or task similarity. Some insights are presented as follows.

**Remark 1.** *Term $F_2^1$ is not always smaller than Term $F_1^1$, indicating instability in their size relationship as the parameters vary. However, as shown in Lemma 10, we find that when $\frac{M_{max}+s}{d} < \frac{1}{T-1}$, this instability is resolved, and the inequality $F_2^1 < F_1^1$ consistently holds. This suggests that with a smaller memory buffer or a larger number of parameters, the model forgets less than without memory when all tasks are highly similar. Both $r^{T-1} - r^{i-1}$ and $u^{T-1} - u^{i-1}$ initially decreases and then increases as $d$ grows, with the corresponding critical points obeying $r = (\frac{i-1}{T-1})^{\frac{1}{T-i}}$ and $u = (\frac{i-1}{T-1})^{\frac{1}{T-i}}$, respectively. Therefore, when we fix $s, i, T$, the critical point w.r.t $d$ of the former will be smaller than that of the latter, indicating that a larger $d$ is more beneficial for the case with a buffer compared to without a buffer, especially when the buffer size is large.*

In Remark 1, we discuss the impact of Term $F_2^1$ on forgetting. We find that for MCL to effectively mitigate forgetting, the memory buffer size, model parameters, and the number of tasks must satisfy the condition $\frac{M_{max}+s}{d} < \frac{1}{T-1}$. Before we discuss the impact of Term $F_2^2$ on forgetting, we consider some fixed number of tasks for Theorem 1 in the following Lemma 2:

**Lemma 2.** *Suppose $d > s + M_{max} + 2$, we have*

$$\mathbb{E}[F_2] = (ur - r)\|\boldsymbol{w}_1^*\|^2 + \frac{s}{d}\|\boldsymbol{w}_2^* - \boldsymbol{w}_1^*\|^2, \tag{15}$$

$$\mathbb{E}[G_2] = \frac{1}{2}\{ur(\|\boldsymbol{w}_1^*\|^2 + \|\boldsymbol{w}_2^*\|^2) + [\frac{M_{max}}{d} + (u+1)\frac{s}{d}]\|\boldsymbol{w}_2^* - \boldsymbol{w}_1^*\|^2\}, \tag{16}$$

$$\mathbb{E}[F_3] = \frac{1}{2}\sum_{i=1}^{2}[(u^2 - u^{i-1})r\|\boldsymbol{w}_i^*\|^2 + \frac{s}{d}\|\boldsymbol{w}_3^* - \boldsymbol{w}_i^*\|^2] + (\frac{uM_{max}}{d} + \frac{u^2 s}{d})\|\boldsymbol{w}_2^* - \boldsymbol{w}_1^*\|^2. \tag{17}$$

When $M_{max} = 0$, Equation (11) is equivalent to Equation (9). Therefore, analyzing the monotonicity of Term $F_2^2$ w.r.t $M_{max}$ will provide a clearer comparison between memory-based methods and those without a memory buffer regarding forgetting.

**Remark 2.** *In Lemma 2, we find that Term $F_2^2 = 0$ when $T = 2$, indicating that the effect of memory on $\mathbb{E}[F_2]$ only depends on the term $ur - r$. This term decreases as $M_{max}$ increases from $0$ to $s$, leading to a reduction in $\mathbb{E}[F_2]$. Therefore, when training with only two tasks, increasing the memory buffer size is beneficial for mitigating forgetting.*

To analyze the impact of Term $F_2^2$ on forgetting, we consider more than two tasks. We present the monotonicity of Term $F_2^2$ w.r.t $M_{max}$ for the case when $T = 3$ in the following Remark 3:

**Remark 3.** *When $T = 3$, the derivative of the term $\frac{uM_{max}}{d} + \frac{u^2 s}{d}$ on $M_{max}$ is given by $\frac{1}{d}[-\frac{M_{max}}{d} - 1 - (\frac{2s}{d} - 1)\frac{M_{max}+s}{d}]$. This derivative is negative under the condition $\frac{M_{max}+s}{d} < \frac{1}{2}$, indicating that the term $\frac{uM_{max}}{d} + \frac{u^2 s}{d}$ decreases as $M_{max}$ increases from $0$ to $s$ under this condition. Furthermore, when $\frac{M_{max}+s}{d} < \frac{1}{2}$ and $T = 3$, a larger memory buffer will result in less forgetting compared to training without memory. However, it is important to note that while a larger memory buffer is beneficial, it requires a correspondingly larger model due to the constraint $\frac{M_{max}+s}{d} < \frac{1}{2}$.*

The discussion above demonstrates that enlarging the memory buffer size will always mitigate forgetting when considering only two tasks. However, for $T = 3$, under the condition $\frac{M_{max}+s}{d} < \frac{1}{2}$, enlarging memory buffer size can be beneficial. The analysis provided by (Lin et al., 2023), which focuses on the effects of various parameters on forgetting and generalization for $T = 2$, is not applicable to our study, as extreme cases begin to arise when $T > 2$.

**Remark 4.** *We find that for $T > 3$, enlarging the memory size is also not always advantageous for mitigating forgetting. To explain this phenomenon, we prefer to use numerical methods. As shown in Figure (1) (a)(b), there are specific combinations of $i$, $j$, and $T$ that cause Term $F_2^2$ w.r.t. $M_{max}$ to increase. This implies that extreme cases may arise: when task $j$ is highly dissimilar to task $i$, while other tasks remain similar to each other, enlarging the memory buffer exacerbates forgetting. Furthermore, we find that $\mathbb{E}[F_T] \to 0$ as $d \to \infty$, indicating that a large-scale model will result in zero forgetting, even though extreme cases may still occur.*

Next, we will consider the impact of memory on the expected generalization error. When $M_{max} = 0$, Equation (12) is equivalent to Equation (10). Therefore, we examine how Equation (12) changes as $M_{max}$ varies from $0$ to $s$, as discussed in Remark 5. This analysis will help us understand how varying memory buffer sizes influence the generalization error.

**Remark 5.** *When $T = 2$, the term $ur$ in Equation 16 decreases as $M_{max}$ increases, indicating that a larger memory buffer improves generalization when two tasks are highly similar. As for the remaining term in Equation 16, taking the derivative of $\frac{M_{max}}{d} + (u+1)\frac{s}{d}$ on $M_{max}$, we have $\frac{1}{d}(1 - \frac{s}{d})$. This shows that, under overparameterized linear models, $\frac{M_{max}}{d} + (u+1)\frac{s}{d}$ increases as $M_{max}$ increases. Hence, a larger memory buffer can hurt generalization when the two tasks are highly dissimilar.*

**Remark 6.** *When $T > 2$, this result changes. Specifically, $G_2^1$ decreases as $M_{max}$ increases, making the effect of $G_2^2$ in Equation 12 more critical for analysis. We observe that enlarging the memory buffer can reduce generalization error in some specific cases, as demonstrated in Figure 1 (c)(d).*

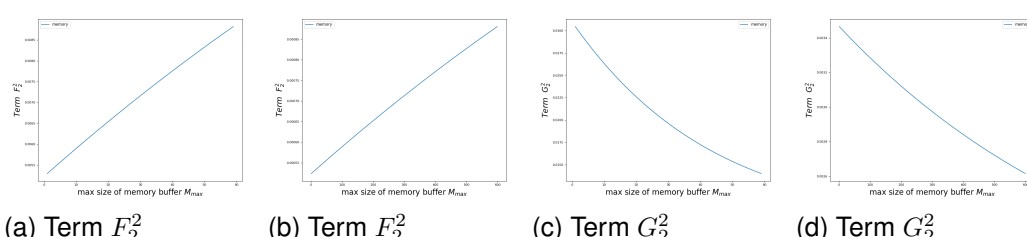

(a) Term $F_2^2$      (b) Term $F_2^2$      (c) Term $G_2^2$      (d) Term $G_2^2$

**Figure 1:** Monotonicity w.r.t. $M_{max}$ for Term $F_2^2$ (a)&(b) and Term $G_2^2$ (c)&(d) under special cases, respectively. Each term is analyzed for both a small-scale model and a large-scale model to illustrate our findings. For the monotonicity of Term $F_2^2$ w.r.t. $M_{max}$, we fix $d = 10000, T = 50, s = 70, i = 20, j = 30$ for Subfigure (a), and $d = 1000000, T = 500, s = 700, i = 200, j = 300$ for Subfigure (b). Under these settings, Term $F_2^2$ increases as $M_{max}$ increases. For the monotonicity of Term $G_2^2$ w.r.t. $M_{max}$, we fix $d = 500, T = 50, j = 40, s = 70$ for Subfigure (c), and $d = 100000, t = 500, j = 400, s = 700$ for Subfigure (d). In both cases, Term $G_2^2$ decreases as $M_{max}$ increases.

*This implies that despite the presence of highly dissimilar tasks, some beneficial cases may arise. For instance, when task $j$ is highly dissimilar to task $i$, as shown in Figure 1 (c)(d), but other tasks remain similar to each other, enlarging the memory buffer can still mitigate forgetting.*

## 4.2 FULL REHEARSAL MEMORY BUFFER

The full rehearsal memory buffer is considered as an unlimited memory buffer, which stores all previous datasets and retrains the model using both the previous and current task datasets. In this section, we also provide explicit expressions for the expected forgetting and generalization errors under overparameterized linear models. To simplify our statements, we define $f(l) := 1 - \frac{ls}{d}$.

**Theorem 2.** *Suppose $d > s + M_{max} + 2$. When $T \geq 2$, we have*

$$
\mathbb{E}[F_T] = \frac{1}{T-1} \bigg\{ \sum_{i=1}^{T-1} \bigg[ \underbrace{\prod_{l=1}^{T} f(l) - \prod_{l=1}^{i} f(l)}_{Term\ F_3^1} \bigg] \|\boldsymbol{w}_i^*\|^2
$$

$$
+ \frac{s}{d} \sum_{j>i}^{T} \bigg[ \underbrace{\bigg( \sum_{k=j}^{T} + \sum_{k=i}^{T} \bigg) \prod_{l=k+1}^{T} f(l) - \sum_{k=i}^{j} \prod_{l=k+1}^{j} f(l)}_{Term\ F_3^2} \bigg] \|\boldsymbol{w}_j^* - \boldsymbol{w}_i^*\|^2 \bigg\}, \tag{18}
$$

$$
\mathbb{E}[G_t] = \frac{1}{T} \sum_{i=1}^{T} \underbrace{\prod_{l=1}^{T} f(l)}_{Term\ G_3^1} \|\boldsymbol{w}_i^*\|^2 + \frac{s}{Td} \sum_{i=1}^{T} \sum_{j=1}^{T} \underbrace{\sum_{k=j}^{T} \prod_{l=k+1}^{T} f(l)}_{Term\ G_3^2} \|\boldsymbol{w}_j^* - \boldsymbol{w}_i^*\|^2. \tag{19}
$$

*Proof Sketch.* First, we perform a detailed matrix decomposition of the data $\hat{\boldsymbol{X}}_t$:

$$
\sum_{k=1}^{t} \sum_{l=(k-1)s+1}^{ks} \boldsymbol{E}_{ts}^l \hat{\boldsymbol{X}}_t^\top \boldsymbol{w}_k^*. \tag{20}
$$

Second, we combine the matrix decomposition of $\hat{\boldsymbol{X}}_t$ from the first step with the properties of the decomposed projector discussed in Appendix A.1 to derive the model error:

$$
\mathbb{E}\|\boldsymbol{w}_t - \boldsymbol{w}_i^*\|^2 = \bigg( 1 - \frac{ts}{d} \bigg) \mathbb{E}\|\boldsymbol{w}_{t-1} - \boldsymbol{w}_i^*\|^2 + \frac{s}{d} \sum_{j=1}^{t} \|\boldsymbol{w}_j^* - \boldsymbol{w}_i^*\|^2. \tag{21}
$$

Finally, we recursively apply Equation (21) and substitute the results into Equation (7) and (8) to obtain explicit expressions for expected forgetting and generalization errors. Then, we finish the proof and leave the details of this proof in Appendix C.1. □

**Multi-task Training** For the $t$-th training round, the full rehearsal memory buffer retains all samples from the previous $t-1$ tasks. Consequently, at any given training round, training with a full rehearsal memory buffer effectively operates as a form of multi-task learning (MTL), utilizing samples from both the previous tasks and the current task simultaneously. Previous theoretical analyses of MTL often rely on concepts such as VC dimension (Crammer & Mansour, 2012; Ben-David & Borbely, 2008), covering number (Baxter, 2000), and Rademacher complexity (Maurer, 2006; Pontil & Maurer, 2013) to derive generalization bounds. However, these approaches do not account for the overparameterized regime and fail to provide explicit forms for generalization error. In Theorem 2, we address these limitations by offering explicit expressions for forgetting and generalization errors in MCL under the overparameterized linear regime. Additionally, while MTL learns tasks simultaneously, MCL handles a sequential stream of tasks.

Training with a full rehearsal buffer may indeed exacerbate forgetting in certain cases, similar to the reservoir sampling-based memory buffer discussed in Subsection 4.1. Moreover, training with the full rehearsal buffer in these scenarios results in significantly higher computational and storage costs. Therefore, we will analyze the impact of memory size and model parameters on the forgetting and generalization errors discussed in Theorem 2. It is important to emphasize that increasing memory size is effectively equivalent to increasing the number of tasks, $T$. Next, we will discuss the coefficients in front of each norm term. Some key insights are presented as follows:

**Remark 7.** *We observe that Term $F_3^1$ is not always smaller than Term $F_1^1$, which is similar to the phenomenon described in Remark 1. However, when $\frac{s}{d} < \frac{1}{T^2}$, the inequality $F_3^1 < F_1^1$ consistently holds, as shown in Appendix C.2. This suggests that having more parameters can help mitigate forgetting when tasks are highly similar. When $d \to \infty$, Term $F_3^1 \to 0$, indicating that a larger model will lead to zero forgetting when tasks are highly similar.*

In Remark 7, we establish the condition $\frac{s}{d} < \frac{1}{T^2}$ to ensure that when tasks are similar, incorporating memory helps mitigate forgetting compared to the no-memory case. Next, we will examine the impact of $F_3^2$ on the expected forgetting.

**Remark 8.** *Based on Equation 18, we have $\mathbb{E}[F_2] = (f(2) - 1)r\|\boldsymbol{w}_1^*\|^2 + \frac{s}{d}\|\boldsymbol{w}_2^* - \boldsymbol{w}_1^*\|^2$. From Equation 9, it follows that $\mathbb{E}[F_2] = (r - 1)r\|\boldsymbol{w}_1^*\|^2 + \frac{s}{d}\|\boldsymbol{w}_2^* - \boldsymbol{w}_1^*\|^2$. Given that $f(2) < r$, we conclude that the expected forgetting of the memory-based method is less than that of the method without memory, indicating that memory-based approaches forget less than those without memory.*

Remark 8 demonstrates that incorporating a memory buffer can mitigate forgetting more effectively than methods without a memory buffer when only two tasks are considered, without any additional conditions. As the number of tasks, $T$, Equation 19 introduces the ground truth $\|\boldsymbol{w}_T^*\|$. Therefore, if the $\{T + 1\}$-th task is similar to all previous tasks, we can ignore the term $\|\boldsymbol{w}_{T+1}^* - \boldsymbol{w}_i^*\|$ for any $i \in [T]$. As for $j \neq T$ and any $i \in [T - 1]$, $F_2^2$ will decrease. Hence, when additional tasks are considered, and the new task is similar to previous ones, storing more samples will lead to reduced forgetting. Moreover, when combined with the condition discussed in Remark 7, we conclude that if the new task is similar to previous tasks and $\frac{s}{d} < \frac{1}{T^2}$, incorporating a memory buffer will mitigate forgetting more effectively than methods without memory. Furthermore, we observe that $\mathbb{E}[F_T]$ in Equation 18 approaches zero as $d \to \infty$, indicating that training with a large-scale model will eventually result in zero forgetting.

Next, we will consider the impact of memory on the expected generalization error. Similarly, we compare the coefficients of each norm in Equation 19 with the method without a memory buffer in Equation 10. This comparison helps us assess how incorporating a memory buffer influences the generalization performance relative to methods that do not utilize memory.

**Remark 9.** *It is evident that $G_3^1 < r^T$ implies that incorporating memory can reduce generalization error when tasks are similar compared to methods without memory. However, we observe that $G_3^2 > 1 > r^{T-i}$ for any $i \in [T]$ and fixed $j$, indicating that when tasks are dissimilar, incorporating a memory buffer results in worse generalization than without memory. Furthermore, the second sum term of Equation 19 increases as $T$ increases, eventually surpassing the second sum term of Equation*

*10. This suggests that storing more samples from previous tasks can lead to worse generalization, particularly when there are highly dissimilar tasks as the number of tasks increases.*

Remark 9 shows that the impact of training with a full rehearsal memory buffer on generalization is highly dependent on task similarity. When all tasks are similar to each other, incorporating full rehearsal results in better generalization compared to training without memory. Conversely, if there are highly dissimilar tasks, incorporating full rehearsal will lead to worse generalization than training without memory. This suggests that task similarity plays a key role in determining the effectiveness of memory buffers for generalization in continual learning.

## 5  CONCLUSION

In this paper, we studied MCL in overparameterized linear models. Specifically, we analyzed the impact of limited-size memory and unlimited-size memory in CL. To achieve this, we selected the reservoir sampling-based memory buffer and the full rehearsal memory buffer as the primary focus of our analysis. We provided explicit expressions for the expected forgetting and generalization errors for both memory strategies. Additionally, we investigated the impact of memory buffer size and the number of model parameters on both forgetting and generalization errors. To verify our findings, we conducted a comprehensive mathematical analysis for scenarios with a small number of tasks and employed numerical analysis for larger task scenarios.

**Limitation:** Our work focuses on overparameterized linear models, without considering deep neural networks (DNNs) or nonlinear activation functions. This simplifies the theoretical analysis and allows us to derive explicit expressions for expected forgetting and generalization errors. However, extending this analysis to DNNs and incorporating the effects of nonlinear activation functions remains an important direction for future work.

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

| *Indices* | |
|---|---|
| $e_n^i$ | The standard basis vector of length $n$, where the $i$-th element is 1 and all other elements are 0. |
| $\boldsymbol{E}_n^i$ | $\mathrm{diag}\{e_n^i\}$ |
| $\mathcal{N}(0,1)$ | Standard Gaussian distribution. |
| $\mathbb{E}[x]$ | The expected value of $x$. |
| *Parameters* | |
| $T$ | Total number of tasks. |
| $s_t$ | Sample size of the $t$-th task. |
| $d$ | The dimension of parameters/feature vectors. |
| *Random Variables* | |
| $\mathcal{D}_t$ | The original dataset for the $t$-th task. |
| $\mathcal{M}_t$ | The memory buffer for the $t$-th task. |
| $\hat{\mathcal{D}}_t$ | The expanded training dataset for the $t$-th task by concatenating $\mathcal{M}_t$ and $\mathcal{D}_t$. |
| $\boldsymbol{X}_t$ | The feature matrix for the $t$-th task. |
| $\boldsymbol{X}_{\mathcal{M}_t}$ | The feature matrix stored in the memory buffer. |
| $\hat{\boldsymbol{X}}_t$ | Concatenation of $\boldsymbol{X}_t$ and $\boldsymbol{X}_{\mathcal{M}_t}$, given by $[\boldsymbol{X}_{\mathcal{M}_t}, \boldsymbol{X}_t]$. |
| $\boldsymbol{y}_t$ | The corresponding label of $\boldsymbol{X}_t$. |
| $\boldsymbol{y}_{\mathcal{M}_t}$ | The corresponding label of $\boldsymbol{X}_{\mathcal{M}_t}$. |
| $\hat{\boldsymbol{y}}_t$ | Concatenation of $\boldsymbol{y}_t$ and $\boldsymbol{y}_{\mathcal{M}_t}$, given by $[\boldsymbol{y}_{\mathcal{M}_t}^\top, \boldsymbol{y}_t^\top]^\top$. |
| $M_{\max}$ | The maximum size of the memory buffer. |
| $M_t$ | The current size of the memory buffer for the $t$-th task. |

**Table 2:** Notation table for key indices, parameters, and random variables used in the paper.

## A USEFUL LEMMAS

### A.1 PROPERTIES OF THE PROJECTOR

For any task $t \in [T]$ and any $l \in [s + M_t]$, we define $\boldsymbol{P}_t := \hat{\boldsymbol{X}}_t(\hat{\boldsymbol{X}}_t^\top \hat{\boldsymbol{X}}_t)^{-1}\hat{\boldsymbol{X}}_t^\top$, $\boldsymbol{P}_t^l := \hat{\boldsymbol{X}}_t(\hat{\boldsymbol{X}}_t^\top \hat{\boldsymbol{X}}_t)^{-1}\boldsymbol{E}_{s+M_t}^l\hat{\boldsymbol{X}}_t^\top$ and $\boldsymbol{p}_t^l := \hat{\boldsymbol{X}}_t(\hat{\boldsymbol{X}}_t^\top \hat{\boldsymbol{X}}_t)^{-1}e_{s+M_t}^l$.

The properties of $\boldsymbol{P}_t$ have been thoroughly discussed in linear algebra, but the characteristics of $\boldsymbol{P}_t^l$ have not been fully explored. Interestingly, $\boldsymbol{P}_t^l$ can serves as a decomposition of $\boldsymbol{P}_t$, meaning that $\boldsymbol{P}_t = \sum_{l=1}^{s+M_t} \boldsymbol{P}_t^l$. In this section, we propose some properties of the projector $\boldsymbol{P}_t^l$ and its projected subspace $\mathcal{P}_t^l$.

**Lemma 3.** *For any task $t \in [T]$ and any $l \in [s + M_t]$, $\boldsymbol{P}_t^l$ is a projector to a 1-dim subspace $\mathcal{P}_t^l$ which is spanned by $\boldsymbol{p}_t^l$.*

*Proof.* By verifying that $(\boldsymbol{P}_t^l)^2 = \boldsymbol{P}_t^l$, we conclude that $\boldsymbol{P}_t^l$ is idempotent and thus a projector. Next, we should confirm the dimension of the projected subspace and identify its basis vectors.

$$dim(\mathcal{P}_t^l) = rank(\boldsymbol{P}_t^l) \leq rank(\boldsymbol{E}_{s+M_t}^l\hat{\boldsymbol{X}}_t^\top) = 1,$$

so the dimension of $\mathcal{P}_t^l$ is 1. We can find the vector in $\mathcal{P}_t^l$ as follows:

$$\boldsymbol{P}_t^l\boldsymbol{p}_t^l = \boldsymbol{p}_t^l.$$

Since the dimension of $\mathcal{P}_t^l$ is 1, we can choose any vector in $\mathcal{P}_t^l$ as its basis vector. Therefore, we select $\boldsymbol{p}_t^l$ as its basis vector $\qquad\square$

In Lemma 3, we verify that any $\boldsymbol{P}_t^l$ is a projector and determine the dimension and basis vectors of its projected subspace $\mathcal{P}_t^l$. Additionally, as shown in the following Lemma 4, we find that applying $\boldsymbol{P}_t^l$ to any vector in $\mathbb{R}^d$ results in a scalar multiple of $\boldsymbol{p}_t^l$.

**Lemma 4.** *For any vector $\boldsymbol{v} \in \mathcal{R}^d$, we have*

$$\boldsymbol{P}_t^l\boldsymbol{v} = c_v\boldsymbol{p}_t^l, \tag{22}$$

*where $c_v$ is a constant.*

Next, we will present the probabilistic properties of any projector $\boldsymbol{P}_t^l$. In the following Lemma 5, we demonstrate that the expectation of the inner product between two different projection operators acting on any two vectors is zero. In Lemma 7, we provide the expectation of the norm of any vector in $\mathcal{R}^d$ when acted upon by any projection operator.

**Lemma 5.** *Suppose $d \geq s + M_t + 2$. For any given vectors $\boldsymbol{v}_1, \boldsymbol{v}_2 \in \mathbb{R}^d$ and $m, n \in [s + M_t]$ where $m \neq n$, we have*

$$\mathbb{E}\langle \boldsymbol{P}_t^m \boldsymbol{v}_1, \boldsymbol{P}_t^n \boldsymbol{v}_2 \rangle = 0. \tag{23}$$

*Proof.* By Remark 4, there are constants $c_1$ and $c_2$ such that

$$\boldsymbol{P}_t^m \boldsymbol{v}_1 = c_1 \boldsymbol{p}_t^m, \qquad \boldsymbol{P}_t^n \boldsymbol{v}_2 = c_2 \boldsymbol{p}_t^n. \tag{24}$$

By substituting Equation (24) back to Equation (23), we have

$$\mathbb{E}\langle \boldsymbol{P}_t^m \boldsymbol{v}_1, \boldsymbol{P}_t^n \boldsymbol{v}_2 \rangle = \mathbb{E}\langle c_1 \boldsymbol{p}_t^m, c_2 \boldsymbol{p}_t^n \rangle = c_1 c_2 (\boldsymbol{e}_{s+M_t}^m)^\top \mathbb{E}((\hat{\boldsymbol{X}}_t^\top \hat{\boldsymbol{X}}_t)^{-1}) \boldsymbol{e}_{s+M_t}^n. \tag{25}$$

We know that $(\hat{\boldsymbol{X}}_t^\top \hat{\boldsymbol{X}}_t)^{-1}$ follows the inverse-Wishart distribution. For $d \geq s + M_t + 2$,

$$\mathbb{E}((\hat{\boldsymbol{X}}_t^\top \hat{\boldsymbol{X}}_t)^{-1}) = \frac{1}{d - s - M_t + 1} \boldsymbol{I}. \tag{26}$$

Due to the condition $m \neq n$, by substituting Equation (26) back to Equation (25), we obtain that

$$\mathbb{E}\langle \boldsymbol{P}_t^m \boldsymbol{v}_1, \boldsymbol{P}_t^n \boldsymbol{v}_2 \rangle = 0.$$

$\square$

**Lemma 6.** *For any task $t$ and any $l \in [s + M_t]$, $\mathcal{P}_t^l$ has rotational symmetry.*

*Proof.* For any rotation $\boldsymbol{S} \in SO(p)$ where $SO(p) \subseteq \mathcal{R}^{d \times d}$ denotes the set of all rotations in $p$-dimensional space, we have

$$\boldsymbol{S}\boldsymbol{p}_t^l = \boldsymbol{S}\hat{\boldsymbol{X}}_t((\boldsymbol{S}\hat{\boldsymbol{X}}_t)^\top (\boldsymbol{S}\hat{\boldsymbol{X}}_t))^{-1} \boldsymbol{e}_l^{s+M_t}. \tag{27}$$

Because of the rotational symmetry of Gaussian distribution, we know that the rotated random matrices $\boldsymbol{S}\hat{\boldsymbol{X}}_t$ has the same probability distribution with the original random matrices $\hat{\boldsymbol{X}}_t$. Therefore, by Eq. 27, we can conclude that $\boldsymbol{S}\boldsymbol{p}_t^l$ has the same probability distribution as $\boldsymbol{p}_t^l$. $\square$

In Lemma 3, we verify that $\mathcal{P}_t^l$ has rotational symmetry for any task $t$ and any $l \in [s + M_t]$. Therefore, we can directly apply the Lemma16 from (Ju et al., 2023) to obtain the following Lemma 7.

**Lemma 7.** *Considering any random projector $\boldsymbol{P}_t^l \in \mathcal{R}^{d \times d}$ to a 1-dim subspace where the subspace has rotational symmetry, then for any given $\boldsymbol{v} \in \mathcal{R}^d$ we must have*

$$\mathbb{E}\|\boldsymbol{P}_t^l \boldsymbol{v}\|_2^2 = \frac{1}{d}\|\boldsymbol{v}\|_2^2.$$

## A.2 RESERVOIR SAMPLING-BASED MEMORY BUFFER

We will introduce Lemma 8, which plays a crucial role in the subsequent proof of the Theorem 1.

**Lemma 8.** *Suppose $d > s + M_{max} + 2$. For any $t \in \{2, \ldots, T\}$ and $i \in [t]$:*

$$\mathbb{E}\|\boldsymbol{w}_t - \boldsymbol{w}_i^*\|^2 = \left(1 - \frac{M_{max} + s}{d}\right)\mathbb{E}\|\boldsymbol{w}_{t-1} - \boldsymbol{w}_i^*\|^2 + \frac{M_{max}}{(t-1)d} \sum_{j=1}^{t-1} \|\boldsymbol{w}_j^* - \boldsymbol{w}_i^*\|^2$$
$$+ \frac{s}{d}\|\boldsymbol{w}_t^* - \boldsymbol{w}_i^*\|^2. \tag{28}$$

*Proof.* We can rewrite Equation (5) as follows:

$$
\begin{aligned}
\boldsymbol{w}_t =& \boldsymbol{w}_{t-1} + \hat{\boldsymbol{X}}_t(\hat{\boldsymbol{X}}_t^\top \hat{\boldsymbol{X}}_t)^{-1}\Bigg(\sum_{k=1}^{t-1}\sum_{l=(k-1)m_{t-1}+1}^{km_{t-1}} \boldsymbol{E}_{M_{max}+s}^l \hat{\boldsymbol{X}}_t^\top \boldsymbol{w}_k^* + \sum_{j=1}^{\bar{m}_{t-1}} \boldsymbol{E}_{M_{max}+s}^{(t-1)m_{t-1}+j} \hat{\boldsymbol{X}}_t^\top \boldsymbol{w}_{t_j}^* \\
& + \sum_{l=M_{max}+1}^{M_{max}+s} \boldsymbol{E}_{M_{max}+s}^l \hat{\boldsymbol{X}}_t^\top \boldsymbol{w}_t^* - \hat{\boldsymbol{X}}_t^\top \boldsymbol{w}_{t-1}\Bigg) \\
=& (\boldsymbol{I} - \boldsymbol{P}_t)\boldsymbol{w}_{t-1} + \sum_{k=1}^{t-1}\sum_{l=(k-1)m_{t-1}+1}^{km_{t-1}} \boldsymbol{P}_t^l \boldsymbol{w}_k^* + \sum_{j=1}^{\bar{m}_{t-1}} \boldsymbol{P}_t^{(t-1)m_{t-1}+j} \boldsymbol{w}_{t_j}^* + \sum_{l=M_{max}+1}^{M_{max}+s} \boldsymbol{P}_t^l \boldsymbol{w}_t^*.
\end{aligned}
\tag{29}
$$

By Equation (29), we have

$$
\begin{aligned}
& \mathbb{E}\|\boldsymbol{w}_t - \boldsymbol{w}_i^*\|^2 \\
=& \mathbb{E}\|(\boldsymbol{I} - \boldsymbol{P}_t)(\boldsymbol{w}_{t-1} - \boldsymbol{w}_i^*) + \sum_{k=1}^{t-1}\sum_{l=(k-1)m_{t-1}+1}^{km_{t-1}} \boldsymbol{P}_t^l \boldsymbol{w}_k^* + \sum_{j=1}^{\bar{m}_{t-1}} \boldsymbol{P}_t^{(t-1)m_{t-1}+j} \boldsymbol{w}_{t_j}^* \\
& + \sum_{l=M_{max}+1}^{M_{max}+s} \boldsymbol{P}_t^l \boldsymbol{w}_t^* - \boldsymbol{P}_t \boldsymbol{w}_i^*\|^2 \\
=& \mathbb{E}\|(\boldsymbol{I} - \boldsymbol{P}_t)(\boldsymbol{w}_{t-1} - \boldsymbol{w}_i^*) + \sum_{k=1}^{t-1}\sum_{l=(k-1)m_{t-1}+1}^{km_{t-1}} \boldsymbol{P}_t^l(\boldsymbol{w}_k^* - \boldsymbol{w}_i^*) + \sum_{j=1}^{\bar{m}_{t-1}} \boldsymbol{P}_t^{(t-1)m_{t-1}+j}(\boldsymbol{w}_{t_j}^* - \boldsymbol{w}_i^*) \\
& + \sum_{l=M_{max}+1}^{M_{max}+s} \boldsymbol{P}_t^l(\boldsymbol{w}_t^* - \boldsymbol{w}_i^*)\|^2 \\
=& \underbrace{\mathbb{E}\|(\boldsymbol{I} - \boldsymbol{P}_t)(\boldsymbol{w}_{t-1} - \boldsymbol{w}_i^*)\|^2}_{(a)} \\
& + \underbrace{\mathbb{E}\|\sum_{k=1}^{t-1}\sum_{l=(k-1)m_{t-1}+1}^{km_{t-1}} \boldsymbol{P}_t^l(\boldsymbol{w}_k^* - \boldsymbol{w}_i^*) + \sum_{j=1}^{\bar{m}_{t-1}} \boldsymbol{P}_t^{(t-1)m_{t-1}+j}(\boldsymbol{w}_{t_j}^* - \boldsymbol{w}_i^*) + \sum_{l=M_{max}+1}^{M_{max}+s} \boldsymbol{P}_t^l(\boldsymbol{w}_t^* - \boldsymbol{w}_i^*)\|^2}_{(b)} \\
& + 2\underbrace{\sum_{k=1}^{t-1}\sum_{l=(k-1)m_{t-1}+1}^{km_{t-1}} \mathbb{E}\langle(\boldsymbol{I} - \boldsymbol{P}_t)(\boldsymbol{w}_{t-1} - \boldsymbol{w}_i^*), \boldsymbol{P}_t^l(\boldsymbol{w}_k^* - \boldsymbol{w}_i^*)\rangle}_{(c)} \\
& + 2\underbrace{\sum_{j=1}^{\bar{m}_{t-1}} \mathbb{E}\langle(\boldsymbol{I} - \boldsymbol{P}_t)(\boldsymbol{w}_{t-1} - \boldsymbol{w}_i^*), \boldsymbol{P}_t^{(t-1)m_{t-1}+j}(\boldsymbol{w}_{t_j}^* - \boldsymbol{w}_i^*)\rangle}_{(d)} \\
& + 2\underbrace{\sum_{l=M_{max}+1}^{M_{max}+s} \mathbb{E}\langle(\boldsymbol{I} - \boldsymbol{P}_t)(\boldsymbol{w}_{t-1} - \boldsymbol{w}_i^*), \boldsymbol{P}_t^l(\boldsymbol{w}_t^* - \boldsymbol{w}_i^*)\rangle}_{(e)}.
\end{aligned}
\tag{30}
$$

(1) For the term (a), we have

$$\mathbb{E}\|(\boldsymbol{I} - \boldsymbol{P}_t)(\boldsymbol{w}_{t-1} - \boldsymbol{w}_i^*)\|^2 = \mathbb{E}\|(\boldsymbol{w}_{t-1} - \boldsymbol{w}_i^*)\|^2 - \mathbb{E}\|\boldsymbol{P}_t(\boldsymbol{w}_{t-1} - \boldsymbol{w}_i^*)\|^2$$

$$= \mathbb{E}\|(\boldsymbol{w}_{t-1} - \boldsymbol{w}_i^*)\|^2 - \frac{M_{max} + s}{d}\mathbb{E}\|(\boldsymbol{w}_{t-1} - \boldsymbol{w}_i^*)\|^2 \qquad (31)$$

$$= (1 - \frac{M_{max} + s}{d})\mathbb{E}\|(\boldsymbol{w}_{t-1} - \boldsymbol{w}_i^*)\|^2,$$

where the first equation is due to properties of orthogonal projection and the second equation is due to the rotational symmetry of the standard normal distribution, which is also described in Lin et al. (2023).

(2) For the term (b), based on Lemma 5 and Lemma 7, we have

$$\mathbb{E}\|\sum_{k=1}^{t-1}\sum_{l=(k-1)m_{t-1}+1}^{km_{t-1}}\boldsymbol{P}_t^l(\boldsymbol{w}_k^* - \boldsymbol{w}_i^*) + \sum_{j=1}^{\bar{m}_{t-1}}\boldsymbol{P}_t^{(t-1)m_{t-1}+j}(\boldsymbol{w}_{t_j}^* - \boldsymbol{w}_i^*) + \sum_{l=M_{max}+1}^{M_{max}+s}\boldsymbol{P}_t^l(\boldsymbol{w}_t^* - \boldsymbol{w}_i^*)\|^2$$

$$\overset{\textcircled{1}}{=} \sum_{k=1}^{t-1}\sum_{l=(k-1)m_{t-1}+1}^{km_{t-1}}\mathbb{E}\|\boldsymbol{P}_t^l(\boldsymbol{w}_k^* - \boldsymbol{w}_i^*)\|^2 + \sum_{j=1}^{\bar{m}_{t-1}}\mathbb{E}\|\boldsymbol{P}_t^{(t-1)m_{t-1}+j}(\boldsymbol{w}_{t_j}^* - \boldsymbol{w}_i^*)\|^2$$

$$+ \sum_{l=M_{max}+1}^{M_{max}+s}\mathbb{E}\|\boldsymbol{P}_t^l(\boldsymbol{w}_t^* - \boldsymbol{w}_i^*)\|^2$$

$$\overset{\textcircled{2}}{=} \sum_{k=1}^{t-1}\frac{m_{t-1}}{d}\|\boldsymbol{w}_k^* - \boldsymbol{w}_i^*\|^2 + \sum_{j=1}^{\bar{m}_{t-1}}\frac{1}{d}\mathbb{E}\|\boldsymbol{w}_{t_j}^* - \boldsymbol{w}_i^*\|^2 + \frac{s}{d}\|\boldsymbol{w}_t^* - \boldsymbol{w}_i^*\|^2$$

$$\overset{\textcircled{3}}{=} \frac{m_{t-1}}{d}\sum_{k=1}^{t-1}\|\boldsymbol{w}_k^* - \boldsymbol{w}_i^*\|^2 + \frac{\bar{m}_{t-1}}{(t-1)d}\sum_{n=1}^{t-1}\|\boldsymbol{w}_n^* - \boldsymbol{w}_i^*\|^2 + \frac{s}{d}\|\boldsymbol{w}_t^* - \boldsymbol{w}_i^*\|^2$$

$$= \left(\frac{(t-1)m_{t-1} + \bar{m}_{t-1}}{(t-1)d}\right)\sum_{k=1}^{t-1}\|\boldsymbol{w}_k^* - \boldsymbol{w}_i^*\|^2 + \frac{s}{d}\|\boldsymbol{w}_t^* - \boldsymbol{w}_i^*\|^2$$

$$= \frac{M_{max}}{(t-1)d}\sum_{k=1}^{t-1}\|\boldsymbol{w}_k^* - \boldsymbol{w}_i^*\|^2 + \frac{s}{d}\|\boldsymbol{w}_t^* - \boldsymbol{w}_i^*\|^2,$$

$$(32)$$

where the equation ① is based on Lemma 5, equation ② is based on Lemma 7, and equation ③ is derived by

$$\mathbb{E}\|\boldsymbol{w}_{t_j}^* - \boldsymbol{w}_i^*\|^2 = \frac{1}{t-1}\sum_{n=1}^{t-1}\|\boldsymbol{w}_n^* - \boldsymbol{w}_i^*\|^2, \qquad (33)$$

which holds for any $j \in [\bar{m}_{t-1}]$.

(3) For the terms (c), (d), and (e), for any $k \in [t]$ and $l \in [M_{max} + s]$, we have

$$\mathbb{E}\langle(\boldsymbol{I} - \boldsymbol{P}_t)(\boldsymbol{w}_{t-1} - \boldsymbol{w}_i^*), \boldsymbol{P}_t^l(\boldsymbol{w}_k^* - \boldsymbol{w}_i^*)\rangle$$

$$= \mathbb{E}[(\boldsymbol{w}_{t-1} - \boldsymbol{w}_i^*)^\top(\boldsymbol{I} - \hat{\boldsymbol{X}}_t(\hat{\boldsymbol{X}}_t^\top\hat{\boldsymbol{X}}_t)^{-1}\hat{\boldsymbol{X}}_t^\top)\hat{\boldsymbol{X}}_t(\hat{\boldsymbol{X}}_t^\top\hat{\boldsymbol{X}}_t)^{-1}\boldsymbol{E}_{M_t+s}^l\hat{\boldsymbol{X}}_t^\top(\boldsymbol{w}_k^* - \boldsymbol{w}_i^*)]. \qquad (34)$$

Through calculation, we obtain that

$$(\boldsymbol{I} - \hat{\boldsymbol{X}}_t(\hat{\boldsymbol{X}}_t^\top\hat{\boldsymbol{X}}_t)^{-1}\hat{\boldsymbol{X}}_t^\top)\hat{\boldsymbol{X}}_t(\hat{\boldsymbol{X}}_t^\top\hat{\boldsymbol{X}}_t)^{-1}\boldsymbol{E}_{M_t+s}^l\hat{\boldsymbol{X}}_t^\top = \boldsymbol{0}, \qquad (35)$$

which leads to

$$(c) = (d) = (e) = 0. \qquad (36)$$

Combining terms (a), (b), (c), (d), and (e), we obtain that

$$\mathbb{E}\|\boldsymbol{w}_t - \boldsymbol{w}_i^*\|^2$$

$$= \left(1 - \frac{M_{max} + s}{d}\right)\mathbb{E}\|(\boldsymbol{w}_{t-1} - \boldsymbol{w}_i^*)\|^2 + \frac{M_{max}}{(t-1)d}\sum_{k=1}^{t-1}\|\boldsymbol{w}_k^* - \boldsymbol{w}_i^*\|^2 + \frac{s}{d}\|\boldsymbol{w}_t^* - \boldsymbol{w}_i^*\|^2 \qquad (37)$$

$$\square$$

### A.3 FULL REHEARSAL MEMORY BUFFER

We will introduce Lemma 9, which plays a crucial role in the subsequent proof of the Theorem 2.

**Lemma 9.** *Suppose $d > s + M_{max} + 2$. For any $t \in \{2, \ldots, T\}$ and $i \in [t]$:*

$$\mathbb{E}\|\boldsymbol{w}_t - \boldsymbol{w}_i^*\|^2 = \left(1 - \frac{ts}{d}\right)\mathbb{E}\|\boldsymbol{w}_{t-1} - \boldsymbol{w}_i^*\|^2 + \frac{s}{d}\sum_{j=1}^{t}\|\boldsymbol{w}_j^* - \boldsymbol{w}_i^*\|^2. \tag{38}$$

*Proof.* We can rewrite Equation (5) as follows:

$$\boldsymbol{w}_t = \boldsymbol{w}_{t-1} + \hat{\boldsymbol{X}}_t(\hat{\boldsymbol{X}}_t^\top \hat{\boldsymbol{X}}_t)^{-1}\left(\sum_{k=1}^{t}\sum_{l=(k-1)s+1}^{ks} \boldsymbol{E}_{ts}^l\hat{\boldsymbol{X}}_t^\top \boldsymbol{w}_k^* - \hat{\boldsymbol{X}}_t^\top \boldsymbol{w}_{t-1}\right)$$

$$= (\boldsymbol{I} - \boldsymbol{P}_t)\boldsymbol{w}_{t-1} + \sum_{k=1}^{t}\sum_{l=(k-1)s+1}^{ks} \boldsymbol{P}_t^l\boldsymbol{w}_k^*. \tag{39}$$

By Equation (39), we have

$$\mathbb{E}\|\boldsymbol{w}_t - \boldsymbol{w}_i^*\|^2$$

$$= \mathbb{E}\|(\boldsymbol{I} - \boldsymbol{P}_t)(\boldsymbol{w}_{t-1} - \boldsymbol{w}_i^*) + \sum_{k=1}^{t}\sum_{l=(k-1)s+1}^{ks} \boldsymbol{P}_t^l\boldsymbol{w}_k^* - \boldsymbol{P}_t\boldsymbol{w}_i^*\|^2$$

$$= \mathbb{E}\|(\boldsymbol{I} - \boldsymbol{P}_t)(\boldsymbol{w}_{t-1} - \boldsymbol{w}_i^*) + \sum_{k=1}^{t}\sum_{l=(k-1)s+1}^{ks} \boldsymbol{P}_t^l(\boldsymbol{w}_k^* - \boldsymbol{w}_i^*)\|^2$$

$$= \underbrace{\mathbb{E}\|(\boldsymbol{I} - \boldsymbol{P}_t)(\boldsymbol{w}_{t-1} - \boldsymbol{w}_i^*)\|^2}_{(a)} + \underbrace{\mathbb{E}\|\sum_{k=1}^{t}\sum_{l=(k-1)s+1}^{ks} \boldsymbol{P}_t^l(\boldsymbol{w}_k^* - \boldsymbol{w}_i^*)\|^2}_{(b)}$$

$$+ \underbrace{2\sum_{k=1}^{t}\sum_{l=(k-1)s+1}^{ks} \mathbb{E}\langle(\boldsymbol{I} - \boldsymbol{P}_t)(\boldsymbol{w}_{t-1} - \boldsymbol{w}_i^*), \boldsymbol{P}_t^l(\boldsymbol{w}_k^* - \boldsymbol{w}_i^*)\rangle}_{(c)}.$$

(1) For the term (a), we have

$$\mathbb{E}\|(\boldsymbol{I} - \boldsymbol{P}_t)(\boldsymbol{w}_{t-1} - \boldsymbol{w}_i^*)\|^2 = \mathbb{E}\|\boldsymbol{w}_{t-1} - \boldsymbol{w}_i^*\|^2 - \mathbb{E}\|\boldsymbol{P}_t(\boldsymbol{w}_{t-1} - \boldsymbol{w}_i^*)\|^2$$
$$= \left(1 - \frac{ts}{d}\right)\mathbb{E}\|\boldsymbol{w}_{t-1} - \boldsymbol{w}_i^*\|^2. \tag{40}$$

(2) For the term (b), based on Lemma 5 and Lemma 7, we have

$$\mathbb{E}\|\sum_{k=1}^{t}\sum_{l=(k-1)s+1}^{ks} \boldsymbol{P}_t^l(\boldsymbol{w}_k^* - \boldsymbol{w}_i^*)\|^2 = \sum_{k=1}^{t}\sum_{l=(k-1)s+1}^{ks} \mathbb{E}\|\boldsymbol{P}_t^l(\boldsymbol{w}_k^* - \boldsymbol{w}_i^*)\|^2$$
$$= \sum_{k=1}^{t}\frac{s}{d}\|\boldsymbol{w}_k^* - \boldsymbol{w}_i^*\|^2. \tag{41}$$

(3) For the term (c) , by applying Equation (35), we have

$$(c) = 0 \tag{42}$$

Combining terms (a), (b), and (c), we obtain that

$$\mathbb{E}\|\boldsymbol{w}_t - \boldsymbol{w}_i^*\|^2 = \left(1 - \frac{ts}{d}\right)\mathbb{E}\|\boldsymbol{w}_{t-1} - \boldsymbol{w}_i^*\|^2 + \sum_{k=1}^{t}\frac{s}{d}\|\boldsymbol{w}_k^* - \boldsymbol{w}_i^*\|^2. \tag{43}$$

$\square$

## B RESERVOIR SAMPLING-BASED MEMORY BUFFER

### B.1 PROOF OF THEOREM 1

By recursively applying Equation (28), we obtain that

$$
\begin{aligned}
&\mathbb{E}\|\boldsymbol{w}_t - \boldsymbol{w}_i^*\|^2 \\
&= \left(1 - \frac{M_{max} + s}{d}\right)^{t-1} \mathbb{E}\|\boldsymbol{w}_1 - \boldsymbol{w}_i^*\|^2 \\
&\quad + \sum_{k=2}^{t} \left(1 - \frac{M_{max} + s}{d}\right)^{t-k} \left[\frac{M_{max}}{(k-1)d} \sum_{j=1}^{k-1} \|\boldsymbol{w}_j^* - \boldsymbol{w}_i^*\|^2 + \frac{s}{d}\|\boldsymbol{w}_k^* - \boldsymbol{w}_i^*\|^2\right].
\end{aligned}
\tag{44}
$$

When $t = 1$, we train without a memory buffer, so we can't directly apply Equation (28). However, we can directly apply the results from Lin et al. (2023). We make the same assumption that $w_0 = 0$, and we obtain that

$$
\begin{aligned}
&\mathbb{E}\|\boldsymbol{w}_t - \boldsymbol{w}_i^*\|^2 \\
&= \left(1 - \frac{M_{max} + s}{d}\right)^{t-1} \left[\left(1 - \frac{s}{d}\right)\|\boldsymbol{w}_i^*\|^2 + \frac{s}{d}\|\boldsymbol{w}_1^* - \boldsymbol{w}_i^*\|\right] \\
&\quad + \sum_{k=2}^{t} \left(1 - \frac{M_{max} + s}{d}\right)^{t-k} \left[\frac{M_{max}}{(k-1)d} \sum_{j=1}^{k-1} \|\boldsymbol{w}_j^* - \boldsymbol{w}_i^*\|^2 + \frac{s}{d}\|\boldsymbol{w}_k^* - \boldsymbol{w}_i^*\|^2\right].
\end{aligned}
\tag{45}
$$

When $t = i$, we can similarly derive that

$$
\begin{aligned}
&\mathbb{E}\|\boldsymbol{w}_i - \boldsymbol{w}_i^*\|^2 \\
&= \left(1 - \frac{M_{max} + s}{d}\right)^{i-1} \left[\left(1 - \frac{s}{d}\right)\|\boldsymbol{w}_i^*\|^2 + \frac{s}{d}\|\boldsymbol{w}_1^* - \boldsymbol{w}_i^*\|\right] \\
&\quad + \sum_{k=2}^{i} \left(1 - \frac{M_{max} + s}{d}\right)^{i-k} \left[\frac{M_{max}}{(k-1)d} \sum_{j=1}^{k-1} \|\boldsymbol{w}_j^* - \boldsymbol{w}_i^*\|^2 + \frac{s}{d}\|\boldsymbol{w}_k^* - \boldsymbol{w}_i^*\|^2\right].
\end{aligned}
\tag{46}
$$

Based on Equation (44) and Equation (46), we calculate the expected forgetting as follows:

$$\mathbb{E}[F_T]$$

$$=\frac{1}{T-1}\sum_{i=1}^{T-1}\mathbb{E}[\|\boldsymbol{w}_T-\boldsymbol{w}_i^*\|^2-\|\boldsymbol{w}_i-\boldsymbol{w}_i^*\|^2]$$

$$=\frac{1}{T-1}\sum_{i=1}^{T-1}\left\{\left[\left(1-\frac{M_{max}+s}{d}\right)^{T-1}-\left(1-\frac{M_{max}+s}{d}\right)^{i-1}\right]\left[\left(1-\frac{s}{d}\right)\|\boldsymbol{w}_i^*\|^2+\frac{s}{d}\|\boldsymbol{w}_1^*-\boldsymbol{w}_i^*\|\right]\right.$$

$$+\sum_{k=2}^{T}\left(1-\frac{M_{max}+s}{d}\right)^{T-k}\left[\frac{M_{max}}{(k-1)d}\sum_{j=1}^{k-1}\|\boldsymbol{w}_j^*-\boldsymbol{w}_i^*\|^2+\frac{s}{d}\|\boldsymbol{w}_k^*-\boldsymbol{w}_i^*\|^2\right]$$

$$\left.-\sum_{k=2}^{i}\left(1-\frac{M_{max}+s}{d}\right)^{i-k}\left[\frac{M_{max}}{(k-1)d}\sum_{j=1}^{k-1}\|\boldsymbol{w}_j^*-\boldsymbol{w}_i^*\|^2+\frac{s}{d}\|\boldsymbol{w}_k^*-\boldsymbol{w}_i^*\|^2\right]\right\}$$

$$=\frac{1}{T-1}\sum_{i=1}^{T-1}\left\{\left[\left(1-\frac{M_{max}+s}{d}\right)^{T-1}-\left(1-\frac{M_{max}+s}{d}\right)^{i-1}\right]\left(1-\frac{s}{d}\right)\|\boldsymbol{w}_i^*\|^2+\frac{s}{d}\|\boldsymbol{w}_T^*-\boldsymbol{w}_i^*\|^2\right.$$

$$+\sum_{j=1}^{T-1}\left[\sum_{k=j+1}^{T}\left(1-\frac{M_{max}+s}{d}\right)^{T-k}\frac{M_{max}}{(k-1)d}+\left(1-\frac{M_{max}+s}{d}\right)^{T-j}\frac{s}{d}\right]\|\boldsymbol{w}_j^*-\boldsymbol{w}_i^*\|^2$$

$$\left.-\sum_{j=1}^{i-1}\left[\sum_{k=j+1}^{i}\left(1-\frac{M_{max}+s}{d}\right)^{i-k}\frac{M_{max}}{(k-1)d}+\left(1-\frac{M_{max}+s}{d}\right)^{i-j}\frac{s}{d}\right]\|\boldsymbol{w}_j^*-\boldsymbol{w}_i^*\|^2\right\}$$

$$=\frac{1}{T-1}\left\{\sum_{i=1}^{T-1}\left\{\left[\left(1-\frac{M_{max}+s}{d}\right)^{T-1}-\left(1-\frac{M_{max}+s}{d}\right)^{i-1}\right]\left(1-\frac{s}{d}\right)\|\boldsymbol{w}_i^*\|^2+\frac{s}{d}\|\boldsymbol{w}_T^*-\boldsymbol{w}_i^*\|^2\right\}\right.$$

$$+\sum_{i<j}^{T-1}\left\{\left(\sum_{k=j+1}^{T}+\sum_{k=i+1}^{T}\right)\left(1-\frac{M_{max}+s}{d}\right)^{T-k}\frac{M_{max}}{(k-1)d}-\sum_{k=i+1}^{j}\left(1-\frac{M_{max}+s}{d}\right)^{j-k}\frac{M_{max}}{(k-1)d}\right.$$

$$\left.\left.+\left[\left(1-\frac{M_{max}+s}{d}\right)^{T-j}+\left(1-\frac{M_{max}+s}{d}\right)^{T-i}-\left(1-\frac{M_{max}+s}{d}\right)^{j-i}\right]\frac{s}{d}\right\}\|\boldsymbol{w}_j^*-\boldsymbol{w}_i^*\|^2\right\}$$

Based on Equation (44), we calculate the expected overall generalization error as follows:

$$\mathbb{E}[G_T]$$

$$=\frac{1}{T}\sum_{i=1}^{T}\mathbb{E}[\|\boldsymbol{w}_T-\boldsymbol{w}_i^*\|^2]$$

$$=\frac{1}{T}\sum_{i=1}^{T}\left(1-\frac{M_{max}+s}{d}\right)^{T-1}\left[\left(1-\frac{s}{d}\right)\|\boldsymbol{w}_i^*\|^2+\frac{s}{d}\|\boldsymbol{w}_1^*-\boldsymbol{w}_i^*\|\right]$$

$$+\frac{1}{T}\sum_{i=1}^{T}\sum_{k=2}^{T}\left(1-\frac{M_{max}+s}{d}\right)^{T-k}\left[\frac{M_{max}}{(k-1)d}\sum_{j=1}^{k-1}\|\boldsymbol{w}_j^*-\boldsymbol{w}_i^*\|^2+\frac{s}{d}\|\boldsymbol{w}_k^*-\boldsymbol{w}_i^*\|^2\right]$$

$$=\frac{1}{T}\sum_{i=1}^{T}\left\{\left[\left(1-\frac{M_{max}+s}{d}\right)^{T-1}\left(1-\frac{s}{d}\right)\|\boldsymbol{w}_i^*\|^2+\frac{s}{d}\|\boldsymbol{w}_T^*-\boldsymbol{w}_i^*\|\right]\right.$$

$$\left.+\sum_{j=1}^{T-1}\left[\sum_{k=j+1}^{T}\left(1-\frac{M_{max}+s}{d}\right)^{T-k}\frac{M_{max}}{(k-1)d}+\left(1-\frac{M_{max}+s}{d}\right)^{T-j}\frac{s}{d}\right]\|\boldsymbol{w}_j^*-\boldsymbol{w}_i^*\|^2\right\}$$

### B.2 ADDITIONAL RESULTS OF THEOREM 1

**Lemma 10.** *For any $i \in [T]$, we have*

$$\begin{cases} u^T - u^i > r^T - r^i & \frac{s}{d} > \frac{1}{i+1} \\ u^T - u^i < r^T - r^i & \frac{M+s}{d} < \frac{1}{T} \end{cases} \tag{47}$$

*Proof.* We first consider the case when $i = T - 1$:

$$u^T - u^{T-1} - r^T - r^{T-1}$$

$$= u^{T-1} \frac{M_{max} + s}{d} + r^{T-1} \frac{s}{d} \tag{48}$$

$$= \frac{1}{d} [s r^{T-1} - (M + s) u^{T-1}]$$

To determine the sign conditions for Equation (48), we introduce the function $f(x) := x(1 - \frac{x}{d})^{T-1}$. We consider the derivative of $f(x)$:

$$f'(x) = (1 - \frac{x}{d})^{T-2} (1 - \frac{Tx}{d}) \tag{49}$$

Therefore, when $x > \frac{d}{T}$, $f'(x) < 0$; when $x < \frac{d}{T}$, $f'(x) > 0$. Furthermore, we have

$$\begin{cases} u^T - u^{T-1} > r^T - r^{T-1} & s > \frac{d}{T} \\ u^T - u^{T-1} < r^T - r^{T-1} & M + s < \frac{d}{T} \end{cases} \tag{50}$$

By reusing Equation (48), we obtain that

$$\begin{cases} u^{i+1} - u^i > r^{i+1} - r^i & s > \frac{d}{i+1} \\ u^{i+1} - u^i < r^{i+1} - r^i & M + s < \frac{d}{i+1} \end{cases} \tag{51}$$

Therefore, by reusing Equation 48 $T - i$ times and summing the expressions, we have

$$\begin{cases} u^T - u^i > r^T - r^i & \frac{s}{d} > \frac{1}{i+1} \\ u^T - u^i < r^T - r^i & \frac{M+s}{d} < \frac{1}{T} \end{cases} \tag{52}$$

$\square$

## C FULL REHEARSAL MEMORY BUFFER

### C.1 PROOF OF THEOREM 2

By recursively applying Equation (38), we obtain that

$$\mathbb{E}\|\boldsymbol{w}_t - \boldsymbol{w}_i^*\|^2$$

$$= \prod_{l=2}^{t} \left(1 - \frac{ls}{d}\right) \|\boldsymbol{w}_1^* - \boldsymbol{w}_i^*\|^2 + \sum_{k=2}^{t} \prod_{l=k+1}^{t} \left(1 - \frac{ls}{d}\right) \frac{s}{d} \sum_{j=1}^{k} \|\boldsymbol{w}_j^* - \boldsymbol{w}_i^*\|^2 \tag{53}$$

$$= \prod_{l=1}^{t} \left(1 - \frac{ls}{d}\right) \|\boldsymbol{w}_i^*\|^2 + \sum_{j=1}^{t} \sum_{k=j}^{t} \prod_{l=k+1}^{t} \left(1 - \frac{ls}{d}\right) \frac{s}{d} \|\boldsymbol{w}_j^* - \boldsymbol{w}_i^*\|^2,$$

where we define the empty product as $\prod_{i=m}^{n} f(i) = 1$ when $m > n$.

When $t = i$, we can similarly derive that

$$\mathbb{E}\|\boldsymbol{w}_i - \boldsymbol{w}_i^*\|^2 = \prod_{l=1}^{i} \left(1 - \frac{ls}{d}\right) \|\boldsymbol{w}_i^*\|^2 + \sum_{j=1}^{i} \sum_{k=j}^{i} \prod_{l=k+1}^{i} \left(1 - \frac{ls}{d}\right) \frac{s}{d} \|\boldsymbol{w}_j^* - \boldsymbol{w}_i^*\|^2. \tag{54}$$

Based on Equation (53) and Equation (54), we calculate the expected forgetting as follows:

$$\mathbb{E}[F_T] = \frac{1}{T-1} \sum_{i=1}^{T-1} \mathbb{E}[\|\boldsymbol{w}_T - \boldsymbol{w}_i^*\|^2 - \|\boldsymbol{w}_i - \boldsymbol{w}_i^*\|^2]$$

$$= \frac{1}{T-1} \sum_{i=1}^{T-1} \left\{ \left[ \prod_{l=1}^{T} \left(1 - \frac{ls}{d}\right) - \prod_{l=1}^{i} \left(1 - \frac{ls}{d}\right) \right] \|\boldsymbol{w}_i^*\|^2 \right.$$

$$\left. + \sum_{j=1}^{T}\sum_{k=j}^{T} \prod_{l=k+1}^{T} \left(1 - \frac{ls}{d}\right) \frac{s}{d} \|\boldsymbol{w}_j^* - \boldsymbol{w}_i^*\|^2 - \sum_{j=1}^{i}\sum_{k=j}^{i} \prod_{l=k+1}^{i} \left(1 - \frac{ls}{d}\right) \frac{s}{d} \|\boldsymbol{w}_j^* - \boldsymbol{w}_i^*\|^2 \right\}$$

$$= \frac{1}{T-1} \left\{ \sum_{i=1}^{T-1} \left[ \prod_{l=1}^{T} \left(1 - \frac{ls}{d}\right) - \prod_{l=1}^{i} \left(1 - \frac{ls}{d}\right) \right] \|\boldsymbol{w}_i^*\|^2 \right.$$

$$\left. + \frac{s}{d} \sum_{j>i}^{T} \left[ \left( \sum_{k=j}^{T} + \sum_{k=i}^{T} \right) \prod_{l=k+1}^{T} \left(1 - \frac{ls}{d}\right) - \sum_{k=i}^{j} \prod_{l=k+1}^{j} \left(1 - \frac{ls}{d}\right) \right] \|\boldsymbol{w}_j^* - \boldsymbol{w}_i^*\|^2 \right\}.$$

Based on Equation (53), we calculate the expected overall generalization error as follows:

$$\mathbb{E}[G_T] = \frac{1}{T} \sum_{i=1}^{T} \mathbb{E}[\|\boldsymbol{w}_T - \boldsymbol{w}_i^*\|^2]$$

$$= \frac{1}{T} \sum_{i=1}^{T} \left\{ \prod_{l=1}^{t} \left(1 - \frac{ls}{d}\right) \|\boldsymbol{w}_i^*\|^2 + \sum_{j=1}^{T}\sum_{k=j}^{T} \prod_{l=k+1}^{T} \left(1 - \frac{ls}{d}\right) \frac{s}{d} \|\boldsymbol{w}_j^* - \boldsymbol{w}_i^*\|^2 \right\}$$

## C.2 ADDITIONAL RESULTS OF THEOREM 2

**Lemma 11.** *For any $i \in [T]$, when $\frac{s}{d} < \frac{1}{T^2}$, we have*

$$\prod_{l=1}^{T} f(l) - \prod_{l=1}^{i} f(l) < f(1)^T - f(1)^i \tag{55}$$

*Proof.* We first consider the case when $i = T - 1$:

$$\prod_{l=1}^{T} f(l) - \prod_{l=1}^{T-1} f(l) - [f(1)^T - f(1)^{T-1}]$$

$$= \prod_{l=1}^{T-1} f(l)[f(T) - 1] - f(1)^{T-1}[f(1) - 1]$$

$$= \frac{s}{d} f(1)^{T-1} - \frac{Ts}{d} \prod_{l=1}^{T-1} f(l) \tag{56}$$

$$= \frac{s}{d} [f(1)^{T-1} - T \prod_{l=1}^{T-1} f(l)]$$

$$= \frac{s}{d} [f(1)^{T-1} - \prod_{l=1}^{T-1} \frac{l+1}{l} f(l)].$$

For any $l \in [T]$, if $\frac{s}{d} < \frac{1}{l^2}$, we have $\frac{l+1}{l} f(l) > f(1)$.

Therefore, if $\frac{s}{d} < \frac{1}{T^2}$, we have that

$$\prod_{l=1}^{T} f(l) - \prod_{l=1}^{T-1} f(l) < f(1)^T - f(1)^{T-1} \tag{57}$$

Based on Eq.(56), we have that

$$\prod_{l=1}^{T-1} f(l) - \prod_{l=1}^{T-2} f(l) < f(1)^{T-1} - f(1)^{T-2},$$

$$\dots \tag{58}$$

$$\prod_{l=1}^{i+1} f(l) - \prod_{l=1}^{i} f(l) < f(1)^{i+1} - f(1)^{i}.$$

By summing the above expressions, we obtain that

$$\prod_{l=1}^{T} f(l) - \prod_{l=1}^{i} f(l) < f(1)^{T} - f(1)^{i}. \tag{59}$$

$\square$