# OpenReview forum: "Towards Understanding Memory buffer based Continual Learning"
_ICLR.cc/2025/Conference — ICLR 2025 Conference Withdrawn Submission_

### Official Review · Reviewer_suz1 · 2024-10-29

**Soundness:** 3
**Presentation:** 2
**Contribution:** 3
**Rating:** 6
**Confidence:** 3

**Summary:**

This paper investigates the effect of memory buffer for linear models in continual learning. It considers two memory buffer settings, and theoretically derives the forgetting and generalization errors under linear models. Based on the theoretical results, the paper argues that a memory buffer can reduces forgetting and improves generalization under certain conditions, while may have the opposite effect under specific model size, task similarity and other factors.

**Strengths:**

1. Analyze the effect of memory buffers in continual learning is an interesting and important problem.
2. Theoretically shows how the memory buffer size effects forgetting and generalization is clear. Meanwhile, analyzing simple cases like $T=2$ helps readers easily understand the relations among them.
3. The paper is well structured.

**Weaknesses:**

1. The paper is not self-contained. The authors should introduce important previous works like [1], for instance, in the appendix or the main paper.
2. For full rehearsal memory buffer setting, the optimization problem is same as the joint (multi-task) training. This is because in the last task $T$, the model is trained on all data and the problem is convex. The paper should discuss the relation between the results in Section 4.2 and joint training.
3. The paper should define "similarity" between tasks formally, which I believe is related to $||w_j^*-w_i^*||_2$. However, in **Remark 1**, it is unclear why the following statement holds true:
> ... This suggests that with a smaller memory buffer or a larger number of parameters, the model forgets less than without memory when all tasks are highly similar.
4. The connection between theories and memory selection is unclear. The paper only discusses one memory selection method: reservoir sampling-based memory buffer, which limits the generalization of findings to other settings.

### Reference
[1] Theory on forgetting and generalization of continual learning. ICML 2023

**Questions:**

1. In Section 3.2, why the memory buffer does not store the labels, and the labels are generated instead. Meanwhile, it is not clear why for reservoir sampling-based memory buffer, the labels are generated by Eq. (1).
2. The paper should have a notation table to sum up all the notations. It is hard to follow the paper since many notations are similar.
3. It is not clear where does the paper get the following conclusion, as stated in the abstract:
> ...  (1) a larger memory buffer must be paired with a larger model to effectively reduce forgetting;

---

> ### Author Response · Authors · 2024-11-25
> **Response to Reviewer suz1**
>
> **Q1.** The paper is not self-contained. The authors should introduce important previous works like [1], for instance, in the appendix or the main paper.
>
> **Response:** Thank you for your suggestion. We clearly state the differences between our work and [1]. Additionally, we summarize the key results from [1] in Lemma 1 to ensure the paper is self-contained.
>
> **Q2.** For full rehearsal memory buffer setting, the optimization problem is same as the joint (multi-task) training. This is because in the last task $T$, the model is trained on all data and the problem is convex. The paper should discuss the relation between the results in Section 4.2 and joint training.
>
> **Response:** Thank you for your suggestion. We have added the relationship between our work and [1] in Section 4.2 for clarity.
>
> **Q3.** The paper should define "similarity" between tasks formally, which I believe is related to $||w_i^*-w_j^*||_2$. However, in Remark 1, it is unclear why the following statement holds true: This suggests that with a smaller memory buffer or a larger number of parameters, the model forgets less than without memory when all tasks are highly similar.
>
> **Response:** The task similarity between task $i$ and task $j$ is characterized  by $||w\_i^*-w\_j^*||\_2$. We have added the definition of task similarity as Definition 1 in Section 4.1.
> In Remark 1, we discuss the impact of $F_2^1$ on forgetting in reservoir sampling-based memory buffers. Compared with the result in Lemma 1, the inequality $F_2^1<F_1^1$ consistently holds under the condition $\frac{M_{max}+s}{d} < \frac{1}{T-1}$. When all tasks are highly similar (i.e., $||w\_i^*-w\_j^*||\_2=0$), $\mathbb{E}[F_T]$ depends primarily on $F_2^1$. The condition $\frac{M_{max}+s}{d} < \frac{1}{T-1}$ suggests that a smaller $M_{max}$ or a larger $d$ will make this inequality hold. Therefore, we arrive at the conclusion: “This suggests that with a smaller memory buffer or a larger number of parameters, the model forgets less than without memory when all tasks are highly similar.”
>
> **Q4.** The connection between theories and memory selection is unclear. The paper only discusses one memory selection method: reservoir sampling-based memory buffer, which limits the generalization of findings to other settings.
>
> **Response:** In this paper, we primarily investigate two types of memory buffers: limited-sized and unlimited-sized. For the limited-sized memory buffer, we adopt the reservoir sampling-based strategy, as described in Section 4.1. For the unlimited-sized memory buffer, we consider the full rehearsal strategy, as outlined in Section 4.2.
>
> In fact, reservoir sampling-based memory buffers can be generalized to other types of memory buffers, where the memory buffer stores $m_{t-1}$ samples for each of the previous $t-1$ tasks. This is a special case of our result, which excludes the random term $\bar{m}_{t-1}$.
>
> **Q5.** In Section 3.2, why the memory buffer does not store the labels, and the labels are generated instead. Meanwhile, it is not clear why for reservoir sampling-based memory buffer, the labels are generated by Eq. (1).
>
> **Response:** We let $X_t$ be the feature matrix and $y_t$ the corresponding label vector for the $t$-th task. We  define $w_i^*$ as the ground truth for the $i$-th task,  which satisfies the equation $y_t=X_t^\top w_i^*$. Therefore, label vector $y_t$ can be expressed in terms of $w_i^*$. To improve clarity, we have rewritten the problem setup in Section 3.1. Therefore, the memory stores both features and labels, where the labels can be expressed in terms of the feature matrix and the ground truth vector.
>
> **Q6.** The paper should have a notation table to sum up all the notations. It is hard to follow the paper since many notations are similar.
>
> **Response:** Thank you for your suggestion. We have provided a notation table in the Appendix for clarity.
>
> **Q7.** It is not clear where does the paper get the following conclusion, as stated in the abstract:(1) a larger memory buffer must be paired with a larger model to effectively reduce forgetting.
>
> **Response:** As described in the response to Q3, the inequality $F_2^1<F_1^1$ consistently holds under the condition $\frac{M_{max}+s}{d} < \frac{1}{T-1}$. Therefore, $d$ should decrease if $M_{max}$ increases to ensure that $\frac{M_{max}+s}{d} < \frac{1}{T-1}$ holds. This implies that a larger memory buffer must be paired with a larger model to effectively reduce forgetting.
>
> **Reference:**
> [1] Theory on forgetting and generalization of continual learning. ICML 2023

---

> > ### Comment · Reviewer_suz1 · 2024-11-26
> >
> > I would like to thanks the authors for the response. All of my questions are answered. I remain positive to this paper.

---

### Official Review · Reviewer_ds34 · 2024-11-01

**Soundness:** 2
**Presentation:** 1
**Contribution:** 2
**Rating:** 3
**Confidence:** 4

**Summary:**

This paper proposes a theoretical analysis of memory-based over-parameterized linear continual learners. More specifically, the authors explicitly express the forgetting and generalization errosr when training a linear model with a MSE and memory mechanism. The authors derive various relation between memory size, dimension of the input and overall forgetting and generalization. Based on this analysis, the authors make various claim regarding the choice of model and memory size to minimize forgetting and maximum generalization.

**Strengths:**

- The motivation of proposing a theoretical framework to give a deeper understanding of memory-based methods is appreciated
- Some limitations have been rightfully addressed

**Weaknesses:**

# Weaknesses
- The introduction is very hard to read and introduces many undefined variables such as $T, s,  M_{max}, d$. The introduced challenges l.48 are unclear; therefore, the context introduction of this theoretical analysis is extremely confusing. I would advise the authors to give a more intuitive understanding of such challenges (i) and (ii). The bullet point contributions at the end of the introduction are appreciated.
- In 3.1, I do not understand the definition of the ground truth vectors $w_i$. Ground truth vectors should be $y_t$ but here they are vectors $w_i$ multiplied to the input to give the ground truth. I imagine that what the authors actually mean is that $w_i$ is the linear learned classifier? However, in that case $y_t$ is the prediction, not the ground truth.
- The definition of reservoir sampling is confusing. Traditionally, the probability of selecting a sample to be put in memory is $\frac{|M|}{k}$ with $|M|$ the memory size and $k$ the stream index. Therefore when the authors write l166 "the probability of any example from the previous $t − 1$ tasks being in the buffer is equal", it does not correspond to reservoir sampling. The probability of being selected decreases over time.
- In 3.1 the diag operator is not defined
- According to 3.1, $X_t^T w_t^*$ is a scalar, however, in equation (1) it is multiplied by $M_{M_{max}}^l$ and the output is a scalar. How is that? Same remark for eq (2)
- Table 1 is never used in the text
- in 3.3, I believe the assumption corresponds to the features. Therefore, each element of $\hat{X}_t$ follows an isotropic Gaussian of mean 0 and variance 1. In that sense, each element of future tasks follows the exact same distribution as previous tasks, but the label is different. This seems unrealistic as in Continual Learning the distribution very likely changes over time from one task to the other.
- I believe the authors consider a classification problem, however the training loss in (3) is most certainly more suited for a regression problem. The authors should also discuss the limitation of studying only one specific loss function.
- In Theorem (1) the authors assume that $d > s + M_{max} +2$. But the value of $M_{max}$ can be very large
- $d$ is sometimes the dimension of the input (section 3.1), sometimes the number of parameters (Table 1). Even if a larger dimension implies more parameters, it would be more interesting to study the impact of over-parameterization, which does not seem to be the case in this paper.
- How much is the overparameterized assumption important here? Could the same analysis apply otherwise? I believe this analysis makes a lot of sense when fine-tuning linear layers of pre-trained models, however this parallel is lacking in the current writing of the paper.
- An experimental validation of the proposed analysis seems necessary to me given the large amount of assumptions made throughout the entire paper.

# Typos
- l.149 lacks the dimension of $x_t$
- In Assumption 1, it should be $t \in [1, T]$

**Questions:**

See weaknesses

---

> ### Author Response · Authors · 2024-11-25
> **Response to Reviewer ds34 (Part 1)**
>
> **Q1.** The introduction is very hard to read and introduces many undefined variables such as $T,s,M_{max},d$. The introduced challenges l.48 are unclear; therefore, the context introduction of this theoretical analysis is extremely confusing. I would advise the authors to give a more intuitive understanding of such challenges (i) and (ii). The bullet point contributions at the end of the introduction are appreciated.
>
> **Response:** We define the variables in the caption of table 1 (line 56). $T$ represents the number of training tasks, $s$ denotes the number of samples per task, $d$ represents the number of training model parameters, and $M_{max}$ is the maximum size of the memory buffer.
>
> **Q2.** In 3.1, I do not understand the definition of the ground truth vectors $w_i$. Ground truth vectors should be $y_t$ but here they are vectors $w_i$  multiplied to the input to give the ground truth. I imagine that what the authors actually mean is that $w_i$  is the linear learned classifier? However, in that case $y_t$ is the prediction, not the ground truth.
>
> **Response:** We let $X_t$ be the feature matrix and $y_t$ the corresponding label vector for the $t$-th task. We  define $w_i^*$ as the ground truth for the $i$-th task,  which satisfies the equation $y_t=X_t^\top w_i^*$. Therefore, label vector $y_t$ can be expressed in terms of $w_i^*$, which is why we refer to $w_i^*$ as the ground truth vector for the $i$-th task. Additionally, in Section 3 of [1], $w_i^*$ is also referred to as the ground truth.
>
> **Q3.** The definition of reservoir sampling is confusing. Traditionally, the probability of selecting a sample to be put in memory is $\frac{|M|}{k}$ wtih $|M|$ the memory size and $k$ the stream index. Therefore when the authors write l166 "the probability of any example from the previous $t-1$ tasks being in the buffer is equal", it does not correspond to reservoir sampling. The probability of being selected decreases over time.
>
> **Response:** "the probability of any example from the previous $t-1$ tasks being in the buffer is equal" means that the probability that any of the examples seen has of being in the buffer is equal to $\frac{|M|}{k}$, as described in Section 3.1 of [1]. Although $\frac{|M|}{k}$ decreases as $k$ increase, the probability of selection remains the same for each example. For instance, if we consider five tasks, the probability of any sample from task 1 being in the buffer is $\frac{|M|}{5}$, and the probability for a sample from task 4 is also $\frac{|M|}{5}$. Therefore, the definition of reservoir sampling used here is consistent.
>
> **Q4.** In 3.1 the diag operator is not defined.
>
> **Response:** The diag operator creates a square diagonal matrix where the elements of the vector become the diagonal entries of the matrix, and all off-diagonal elements are zero.
>
> **Q5.** According to 3.1, $X_t^\top w_t^*$ is a scalar, however, in equation (1) it is multiplied by $M_{M_{max}}^l$ and the output is a scalar. How is that? Same remark for eq (2).
>
> **Response:** We define $X_t$ as a $d\times s_t$ matrix and $w_t^*$ as a $d$-dimensional vector. Consequently, $X_t^\top w_t^*$ is an $s_t$-dimensional vector.  Additionally, the output of Equation (1) is a vector.
>
> **Q6.** Table 1 is never used in the text
>
> **Response:** Table 1 is included in the introduction to facilitate a convenient comparison between our results and those in [1].
>
> **Q7.** In 3.3, I believe the assumption corresponds to the features. Therefore, each element of $\hat X_t$ follows an isotropic Gaussian of mean 0 and variance 1. In that sense, each element of future tasks follows the exact same distribution as previous tasks, but the label is different. This seems unrealistic as in Continual Learning the distribution very likely changes over time from one task to the other.
>
> **Response:** Firstly， we assume $\hat X_t$ follows an isotropic Gaussian to simplify our analysis. This assumption is made for tractability and is a common choice in machine learning, as it provides a general framework. If we were to consider distributions other than the Gaussian for each task, it would introduce significantly more complex coefficients in Eq. (29). Secondly, our results are compared with those in [1], and therefore, we align with Assumption 3.2 from [1]. This alignment allows us to highlight the advantages of training with a memory buffer as compared to no memory in some aspects, as demonstrated in [1]. Finally，although the distribution of each task follows an isotropic Gaussian, the vector $w_i^*$ captures the variation between tasks. The right-hand-side of Eq (11, 12, 18, 19) also includes $||w_i^*-w_j^*||$, which effectively quantifies the difference between two tasks. We also discuss the impact of memory under different task similarity scenarios in the remark.

---

> > ### Author Response · Authors · 2024-11-25
> > **Response to Reviewer ds34 (Part 2)**
> >
> > **Q8.** I believe the authors consider a classification problem, however the training loss in (3) is most certainly more suited for a regression problem. The authors should also discuss the limitation of studying only one specific loss function.
> >
> > **Response:** In this paper, we consider the linear regression problem. Therefore, we use MSE loss as our training loss.
> >
> > **Q9.** In Theorem (1) the authors assume that $d>s+M_{max}+2$. But the value of $M_{max}$ can be very large
> >
> > **Response:** In Theorem (1), we consider the reservoir sampling-based memory buffer. We limit the buffer capacity to not exceed $s$, as described in Line 169.
> >
> > **Q10.** $d$  is sometimes the dimension of the input (section 3.1), sometimes the number of parameters (Table 1). Even if a larger dimension implies more parameters, it would be more interesting to study the impact of over-parameterization, which does not seem to be the case in this paper.
> >
> > **Response:** In this paper, we analyze continual learning under an over-parameterized linear model, i.e. $y=X^\top w$. We assume that the feature matrix $X\in\mathbb R^{d\times s_t}$. Therefore, $d$ is not only the diemension of the input, but also the dimension of parameters $w$ (i.e. the number of parameters).
> >
> > **Q11.** How much is the overparameterized assumption important here? Could the same analysis apply otherwise? I believe this analysis makes a lot of sense when fine-tuning linear layers of pre-trained models, however this parallel is lacking in the current writing of the paper.
> >
> > **Response:** Firstly, 'double descent' is a well-known phenomenon in machine learning, where overparameterized models often achieve lower test error than underparameterized models. This makes the overparameterized regime particularly valuable for further study. Secondly, in the underparameterized regime, there is a unique solution that minimizes the MSE loss (i.e. Eq. (3)), given by $w_t=(\hat X_t^\top\hat X_t)^{-1}\hat X_ty_t$. This solution can then be substituted into Eq. (7) and Eq. (8)  to easily derive the expected forgetting and overall generalization error.  However, in the overparameterized regime, minimizing Eq. (3) results in infinitely many solutions that achieve zero error. Among these solutions, we are particularly interested in the one corresponding to the convergent point of SGD. Therefore, the analysis in the overparameterized regime is more complex than in the underparameterized regime, which is why we focus on the overparameterized case.
> >
> > **Reference:**
> > [1] Theory on forgetting and generalization of continual learning. ICML 2023

---

> ### Comment · Reviewer_ds34 · 2024-11-25
>
> I thank the authors for taking the time to respond to my comments, which have clarified some of my concerns. However, I still believe that this work demonstrates a lot of weaknesses. Notably, the authors ignored the questions regarding the absence of experiments, despite being raised by several reviewers. For these reasons, **I will keep my score unchanged**.
>
> **Q1**: So this is normal for you to use notations in the introduction that are defined in the caption of a table that is never called in the text?
>
> **Q2**: Reviewer **4mXQ** had similar comments on this regard and I really think you should reconsider the writing of this; a set of weights is not a ground truth.
>
> **Q6**: Every introduced table or figure should be called in the text when you write a paper.
>
> **Q8**: I understand, but this is not stated properly in the paper, and it should be, as it is a clear limitation.
>
> **Q10**: Same as before, you should clarify your mathematical notations here. Your claim that $d$ can interchangeably be a dimension or the number of parameters is not obvious in the current writing.

---

### Official Review · Reviewer_sdbe · 2024-11-03

**Soundness:** 2
**Presentation:** 2
**Contribution:** 2
**Rating:** 3
**Confidence:** 4

**Summary:**

This paper provides a theoretical investigation of replay based CL where each task is an overparameterized linear regression task, under two different scenarios, i.e., reservior sampling and a full replay where all previous data are stored. By deriving explicit forms of expected forgetting and generalization errors, the authors analyze the impact of memory size and model size on forgetting and generalization error, under the coupling with task similarity.

**Strengths:**

1. The theory for replayed based CL is very limited, and this paper provides the first explicit forms of forgetting and generalization error for two replay strategies.

2. Analysis are provided to understand the impact of memory.

**Weaknesses:**

1. No experimental studies are provided to justify the theoretical results. In particular, it is not clear if similar phenomenons can be observed in practice with neural networks and real datasets, which challenges the usefulness and importance of the theoretical results in this paper. And it is not clear how the theoretical results can help in practice.

2. It is not convincing that analyzing the full replay case is important here, as this replay strategy is barely used in practice.

3. The presentation needs to be improved. For example, it is not clear what the distinct tasks are in line 172. Figure 1 also needs to be more clear. The last sentence in Remark 6 also seems questionable, as these two figures are talking about generalization error instead of forgetting.

**Questions:**

Besides the weakness above, I have some further questions:

1. In the replay buffer, why not put samples for the same old task together? This may be convenient for analysis, but seems not standard.

2. How did you handle the correlation between the replay samples and the model? For example, data stored in the replay buffer for task $t-1$ have already been seen during the training of $w_{t-1}$. When using this data to update the model at task $t$, the model $w$ should be corrected with $\hat{D}_t$. Address this challenge is important to analyze replay-based CL, but I didn't see how this was particularly addressed in the paper.

3. In section 4.1, the claims about the impact of memory size on forgetting are made only based on $F_2^2$, which seems not very rigorous. The reason is that, when changing the memory, from Theorem 1, most time the coefficient of $F_2^1$ will also change. And unlink the second that depends on task similarity, this first term cannot be very small.

---

> ### Author Response · Authors · 2024-11-25
> **Response to Reviewer sdbe**
>
> **Q1.** It is not convincing that analyzing the full replay case is important here, as this replay strategy is barely used in practice.
>
> **Response:** In our paper, we theoretically analyze the impact of memory on training by comparing two cases: training with a limited-sized memory buffer and with an unlimited-sized memory buffer. Full rehearsal is a typical example of an unlimited-sized memory buffer. Therefore, we provide a theoretical analysis of its effects on both forgetting and generalization.
>
> **Q2.** The presentation needs to be improved. For example, it is not clear what the distinct tasks are in line 172. Figure 1 also needs to be more clear. The last sentence in Remark 6 also seems questionable, as these two figures are talking about generalization error instead of forgetting.
>
> **Response:** "the $m$ distinct tasks from the previous $t-1$ tasks" means that we randomly select $m$ tasks from the set of $t-1$ tasks without replacement. Therefore, any two tasks among the $m$ selected tasks will not overlap. Thank you very much for pointing out the error in the last sentence of Remark 6. We have now corrected it accordingly.
>
>
> **Q3.** In the replay buffer, why not put samples for the same old task together? This may be convenient for analysis, but seems not standard.
>
> **Response:** In this paper, storing samples from the same old task is a special case of our result. "The same old task" implies that the ground truths of these samples are identical. Therefore, the term $||w_j^*-w_i^*||=0$ holds.
>
> **Q4.** How did you handle the correlation between the replay samples and the model? For example, data stored in the replay buffer for task $t-1$ have already been seen during the training of  $w_{t-1}$. When using this data to update the model at task $t$, the model $w$ should be corrected with $\hat D_t$. Address this challenge is important to analyze replay-based CL, but I didn't see how this was particularly addressed in the paper.
>
> **Response:** The convergence point of SGD for minimizing the training loss (MSE) corresponds to the solution of the following optimization problem: $\min_w ||w-w_t||  s.t.\hat X_t^\top w=\hat y_t$. Using the method of Lagrange multipliers, the solution to this problem can be directly obtained as: $w_t=w_{t-1}+\hat X_t(\hat X_t^\top \hat X_t)^{-1}(\hat y_t-\hat X_t^\top w_{t-1})$, where $\hat X_t$ includes the samples from both the $t$-th task and memory buffer, as described in Section 3.3. This formulation accounts for the correlation between the replay samples and the model. Additionally, we provide a more detailed derivation in Eq. (29) in Appendix A.2.
>
>
> **Q5.** In section 4.1, the claims about the impact of memory size on forgetting are made only based on $F_2^2$, which seems not very rigorous. The reason is that, when changing the memory, from Theorem 1, most time the coefficient of $F_2^1$ will also change. And unlink the second that depends on task similarity, this first term cannot be very small.
>
> **Response:** We analyze the impact of memory size on forgetting based on both $F_2^1$ and $F_2^2$. In Remark 1, by comparing $F_2^1$ and $F_2^2$, we conclude that $F_2^1<F_1^1$ always holds when $\frac{M_{max}+s}{d}<\frac{1}{T-1}$. Under this condition, in Remarks 2, 3, and 4, we explore the conditions under which $F_2^2<c_{i,j}$. By combining the insights from Remarks 1, 2, 3, and 4, we can determine under which conditions training with a memory buffer results in less forgetting compared to training with no memory buffer. Therefore, when considering the impact of memory size on forgetting, we account for both $F_2^1$ and $F_2^2$.

---

> > ### Comment · Reviewer_sdbe · 2024-11-27
> >
> > Thanks the authors for the rebuttal and clarifying some of my questions. But my main concerns regarding the weakness still remain: 1) lack of experimental verifications; 2) I understand that full replay is a special case of unlimited memory, but I am still not convinced that analyzing this case is meaningful for standard CL, because this is far from what has been used in practice. In terms of Q4, I understand how the closed form is obtained, but analyzing the forgetting based on these closed form solutions can be challenging due to the sample-model correlation, i.e., $\hat{X}$ is strongly correlated with $w_{t-1}$, which however has not been clarified. Given these, I decide to maintain my score.

---

### Official Review · Reviewer_4mXQ · 2024-11-08

**Soundness:** 1
**Presentation:** 1
**Contribution:** 1
**Rating:** 3
**Confidence:** 2

**Summary:**

The paper seeks to theoretically analyze the impact of memory on continual learning in overparameterized linear models. To do this, the paper starts from error terms for forgetting and generalization defined in the ICML 2023 paper by Lin et al. for continual learning without a memory buffer. The present paper then extends these error terms for continual learning methods that use one of two memory buffers: 1) a buffer created using reservoir sampling (a partial rehearsal buffer) or 2) a buffer that stores all previous samples (a full rehearsal buffer).
The paper claims that these error terms lead to three conclusions:
1) “A larger memory buffer must be paired with a larger model to reduce forgetting effectively”;
2) “A sufficiently large model can lead to zero forgetting”;
3) “A larger memory buffer may improve generalization when tasks are highly similar but can degrade generalization when tasks are highly dissimilar.”

**Strengths:**

**Originality:** Theoretically analyzing the error bounds for forgetting and generalization in continual learning (overparameterized linear) models that use a memory buffer is novel. To the best of my knowledge, there do not currently exist any studies that examine this problem. The most related work is the ICML 2023 paper by Lin et al.

**Significance:** It is useful to study these error bounds for continual learning models that use memory buffers since this has become a predominant strategy for mitigating catastrophic forgetting in experimental works in the field. Theoretical bounds pave the way for a better understanding of experimental findings in the field. Although the error bounds are only derived for overparameterized linear models, future work could explore these bounds for other models like deep neural networks or non-linear models more broadly.

**Quality & Clarity:** While providing theoretical error bounds for the forgetting and generalization of memory-based continual learning methods is useful, the paper is lacking clarity and is difficult to follow. For example, there are several design choices that do not have clear justifications and the mathematical notation is inconsistent, which is expanded upon in the “Weaknesses” section below.

**Weaknesses:**

My biggest issue with the paper is the lack of clarity, which makes it difficult to evaluate the correctness and impact of the presented results. I will expand on several points related to this next.

- It is unclear what the meaning of a “task” is in the paper. The data vectors are said to be sampled from Normal(0,1), but then what is changing from one task to another? This is important for understanding the results. Moreover, the notion of “task similarity” is mentioned several times, including in the concluding findings from the theoretical analysis. However, “task similarity” is never mathematically defined. Are we assuming some bound on the difference between two matrices as their “similarity”? If so, this is not mentioned anywhere in the paper and remains unclear.

- The paper could benefit from more consistent mathematical notation. For example, in Section 3.1, W_T is defined as the set of linear ground truth vectors of all T tasks, but then the vectors of W_T are used to produce the output y=X^T w_t. However, y appears to be a prediction, not a ground truth, so this is unclear. Moreover, in Section 3.2.1, it is stated that “the memory buffer stores m_{t-1} samples” and also that we have “\hat{m}_{t-1}” tasks. The inconsistency of variables makes the math extremely difficult to follow. These are just a few examples.

- Another benefit could come from plain English explanations of theoretical results. The Remarks in the paper are useful for understanding the theoretical results, however, they are written in terms of math and it is unclear what their implications are for the study. These plain English explanations could help the reader better understand the lemmas, theorems, and remarks in the context of the broader continual learning field.

- There are several justifications for design choices missing. For example, there is an assumption that the number of samples for each task is equal (Assumption 1). Is this just to simplify notation? How do we know these results will hold for the more general case when each task does not contain the same number of samples? In Section 3.2.1, it is stated that “When t \geq 2, the memory buffer stores m_{t-1} samples for each of the previous t-1 tasks and 1 sample for each of the \hat{m}_{t-1} distinct tasks from the previous t-1 tasks.” Does this mean the buffer only stores 1 sample per task? If so, this is a very strong assumption and it is unclear why this was chosen.

In the study by Lin et al., there were empirical results with deep neural networks that demonstrated similar findings to the theoretical analyses performed for the overparameterized linear model. This was powerful to demonstrate that their theoretical results might hold for more complex models (like non-linear neural networks). It would be helpful to see a similar type of experimental analysis in the present paper to strengthen the theoretical findings.

**Questions:**

1. How is a “task” being mathematically defined? Why was this specific definition chosen?
2. How is “task similarity” being defined? Why was this specific definition chosen?
3. It would be helpful to see experimental results similar to those with a deep neural network from Lin et al. to enhance the theoretical findings in the present paper.
4. It would be helpful to have plain English explanations of the “Remarks” throughout the paper to better understand the findings.
5. It would be helpful to have more context about the study from Lin et al. so that the reader does not have to read excerpts from Lin et al.’s paper to understand the present paper.
6. In the Introduction, it is mentioned that the challenges for the present study consist of “the coupling of the data matrix and label vector”. What exactly does this mean?
7. What exactly was the setup used to produce the plots in Figure 1? Were Gaussians sampled multiple times to create different “tasks” and then the results averaged for the plots?

---

> ### Author Response · Authors · 2024-11-25
> **Response to Reviewer 4mXQ (Part 1)**
>
> **Q1.**  How is a “task” being mathematically defined? Why was this specific definition chosen?
>
> **Response:** We let $X_t$ be the feature matrix and $y_t$ the corresponding label vector for the $t$-th task. We  define $w_i^*$ as the ground truth for the $i$-th task,  which satisfies the equation $y_t=X_t^\top w_i^*$. Thus, $w_i^*$ characterizes the differences between tasks. To simplify our analysis, we assume $X_t$ follows $\mathcal N(0,1)$ for all $t\in [T]$. Without this assumption, additional parameters would be introduced, leading to significantly more complex results. This assumption is also consistent with Assumption 3.2 of [1].
>
> **Q2.** How is “task similarity” being defined? Why was this specific definition chosen?
>
> **Response:** We define the task similarity between any task $i$ and $j$ as $||w_i^*-w_j^*||^2$. $w_i^*$ serves as a representative of task $i$, allowing us to use the norm of their difference to characterize the similarity between two tasks. This approach is a common setting in linear regression and is consistent with Section 4 in [1]. We have added the definition of task similarity as Definition 1 in Section 4.1.
>
>
> **Q3.** It would be helpful to have more context about the study from Lin et al. so that the reader does not have to read excerpts from Lin et al.’s paper to understand the present paper.
>
> **Response:** Thank you for your suggestion. We clearly state the differences between our work and [1]. Additionally, we summarize the key results from [1] in Lemma 1 to ensure the paper is self-contained.
>
> **Q4.** In the Introduction, it is mentioned that the challenges for the present study consist of “the coupling of the data matrix and label vector”. What exactly does this mean?
>
> **Response:** As stated in the response to Q1, $w_i^*$ satisfies the equation $y_t=X_t^\top w_i^*$. However, our training dataset (Section 3.3) includes both the original dataset and the memory buffer, meaning the expanded feature matrix $\hat X_t$  contains samples from multiple tasks. Consequently, it is not possible to directly define a single $w^*$ for  $\hat X_t$ and $\hat y_t$. To address this issue, we decouple the matrix $\hat X_t$ into a sum of matrices, where each decoupled matrix includes only the samples from a single task, as described in Eq. (1) and Eq. (2).
>
> **Q5.** What exactly was the setup used to produce the plots in Figure 1? Were Gaussians sampled multiple times to create different “tasks” and then the results averaged for the plots?
>
> **Response:** In Figure 1, we select specific values for $d$, $T$, $s$, $i$, and $j$ to calculate Term $F_2^2$ and Term $G_2^2$. Based on the results, When $T$ increases, there may be some extreme situations, including:
> (1) Enlarging the memory size is not always advantageous for mitigating forgetting (as noted in Remark 4).
> (2) Enlarging the memory buffer can reduce the generalization error in some specific cases, even for highly dissimilar tasks (as noted in Remark 6).
>
> **Q6.** The paper could benefit from more consistent mathematical notation. For example, in Section 3.1, $W_T$ is defined as the set of linear ground truth vectors of all T tasks, but then the vectors of $W_T$ are used to produce the output $y=X^T w_t$. However, y appears to be a prediction, not a ground truth, so this is unclear. Moreover, in Section 3.2.1, it is stated that “the memory buffer stores $m_{t-1}$ samples” and also that we have $\hat{m}_{t-1}$ tasks. The inconsistency of variables makes the math extremely difficult to follow. These are just a few examples.
>
> **Response:**  Firstly, in our setting, we refer to the vector in $w^*$ as the ground truth, not $y$, as described in the response to Q1. Secondly, in Section 3.2.1, the number of samples stored in the memory buffer for each previous task is nearly equal. We define the maximum buffer size as $M_{\text{max}}$. Therefore, for task $t$, the memory buffer stores approximately $M_{\text{max}}/ (t-1)$ samples from each of the previous $t-1$ tasks. However, $M_{\text{max}} / (t-1)$ is not always an integer as $t$ changes. To address this, we set  $m_{t-1} = \text{int}\left(M_{\text{max}} / (t-1)\right)$ and $\bar{m}\_{t-1} = M_{\text{max}} - (t-1)m_{t-1}$,  such that  $m\_{t-1}(t-1) + \bar{m}\_{t-1} = M_{\text{max}}$. Thus, the buffer stores $m\_{t-1}$ samples for each of the $t-1$ previous tasks, with a remainder of $\bar{m}\_{t-1}$ samples. We first select $\bar{m}\_{t-1}$ tasks and store one additional sample from each in the buffer. This ensures that the difference in the number of samples stored per task does not exceed 1.

---

> ### Author Response · Authors · 2024-11-25
> **Response to Reviewer 4mXQ (Part 2)**
>
> **Q7.** There are several justifications for design choices missing. For example, there is an assumption that the number of samples for each task is equal (Assumption 1). Is this just to simplify notation? How do we know these results will hold for the more general case when each task does not contain the same number of samples? In Section 3.2.1, it is stated that “When $t \geq 2$, the memory buffer stores $m\_{t-1}$ samples for each of the previous t-1 tasks and 1 sample for each of the $\hat{m}_{t-1}$ distinct tasks from the previous t-1 tasks.” Does this mean the buffer only stores 1 sample per task? If so, this is a very strong assumption and it is unclear why this was chosen.
>
> **Response:** As shown in the response to Q7, the buffer stores at least $m\_{t-1}$ samples per task, not 1. This assumption is consistent with Assumption 3.3 in [1].
>
> **Reference:**
> [1] Theory on forgetting and generalization of continual learning. ICML 2023

---

> > ### Comment · Reviewer_4mXQ · 2024-11-26
> > **Official Comment from Reviewer 4mXQ**
> >
> > Thank you to the authors for the responses to my questions and questions from other reviewers. However, several of my concerns are still not fully addressed (see below) and there remain some issues regarding the clarity of the method and its presentation in the paper. Moreover, I still think the paper could benefit from experimental validation (similar to Lin et al.) as suggested by myself, along with Reviewer sdbe and Reviewer ds34. For these reasons, I am maintaining my original score.
> >
> > **Q1.** I am still confused about the definition of a task. It seems that every "task" follows a normal distribution, so I am confused about what the non-stationarity is concerning data X. What changes in the data from one task to the next? Moreover, similar to Reviewer ds34, I find defining weights as "ground truth" confusing.
> >
> > **Q2.** Similar to Q1, it also seems weird to define task similarity by the similarity of weights. Shouldn't the similarity of tasks also consider the data X?
> >
> > **Q3.** While the current paper does state some of the differences with the ICML study by Lin et al., the present paper could still benefit from more details about the setup and assumptions from Lin et al. I found myself having to reference the ICML paper several times to better understand the current paper.
> >
> > **Q4.** Thank you for the clarification. The paper could benefit from a more expanded description of this "coupling". However, I am still confused about why the data X needs to be decoupled by task? Also is it assumed that we have task labels to know how to decouple X by task? If so, this assumption is not obvious to the reader. Moreover, this goes back to Q1 about the definition of a task as it relates to the data X.
> >
> > **Q6.** My comment was more of a suggestion to improve the notation in the paper for clarity.

---

### Note · Authors · 2025-01-23

I have read and agree with the venue's withdrawal policy on behalf of myself and my co-authors.